

# Cancellous bone and theropod dinosaur locomotion. Part II—a new approach to inferring posture and locomotor biomechanics in extinct tetrapod vertebrates

Peter J. Bishop[1,2,3,4], Scott A. Hocknull[1,2,5], Christofer J. Clemente[6,7], John R. Hutchinson[8], Rod S. Barrett[2,3] and David G. Lloyd[2,3]

[1] Geosciences Program, Queensland Museum, Brisbane, QLD, Australia
[2] School of Allied Health Sciences, Griffith University, Gold Coast, QLD, Australia
[3] Gold Coast Orthopaedic Research, Engineering and Education Alliance, Menzies Health Institute Queensland, Gold Coast, QLD, Australia
[4] Current affiliation: Structure and Motion Laboratory, Department of Comparative Biomedical Sciences, Royal Veterinary College, Hatfield, Hertfordshire, UK
[5] School of Biosciences, University of Melbourne, Melbourne, VIC, Australia
[6] School of Science and Engineering, University of the Sunshine Coast, Maroochydore, QLD, Australia
[7] School of Biological Sciences, University of Queensland, Brisbane, QLD, Australia
[8] Structure and Motion Laboratory, Department of Comparative Biomedical Sciences, Royal Veterinary College, Hatfield, Hertfordshire, UK

Corresponding author
Peter J. Bishop, pbishop@rvc.ac.uk

## ABSTRACT

This paper is the second of a three-part series that investigates the architecture of cancellous bone in the main hindlimb bones of theropod dinosaurs, and uses cancellous bone architectural patterns to infer locomotor biomechanics in extinct non-avian species. Cancellous bone is widely known to be highly sensitive to its mechanical environment, and therefore has the potential to provide insight into locomotor biomechanics in extinct tetrapod vertebrates such as dinosaurs. Here in Part II, a new biomechanical modelling approach is outlined, one which mechanistically links cancellous bone architectural patterns with three-dimensional musculoskeletal and finite element modelling of the hindlimb. In particular, the architecture of cancellous bone is used to derive a single 'characteristic posture' for a given species—one in which bone continuum-level principal stresses best align with cancellous bone fabric—and thereby clarify hindlimb locomotor biomechanics. The quasi-static approach was validated for an extant theropod, the chicken, and is shown to provide a good estimate of limb posture at around mid-stance. It also provides reasonable predictions of bone loading mechanics, especially for the proximal hindlimb, and also provides a broadly accurate assessment of muscle recruitment insofar as limb stabilization is concerned. In addition to being useful for better understanding locomotor biomechanics in extant species, the approach hence provides a new avenue by which to analyse, test and refine palaeobiomechanical hypotheses, not just for extinct theropods, but potentially many other extinct tetrapod groups as well.

## INTRODUCTION

Cancellous bone is highly sensitive and able to adapt its three-dimensional (3D) architecture to its prevailing mechanical environment, such that the overall architecture strongly reflects the loads experienced by whole bones. Cancellous bone architecture can also change when loading conditions change, and this structural alteration takes place in a predictable fashion (*Adachi et al., 2001*; *Barak, Lieberman & Hublin, 2011*; *Biewener et al., 1996*; *Goldstein et al., 1991*; *Guldman et al., 1997*; *Huiskes et al., 2000*; *Mullender & Huiskes, 1995*; *Polk, Blumenfeld & Ahluwalia, 2008*; *Pontzer et al., 2006*; *Radin et al., 1982*; *Richmond et al., 2005*; *Ruimerman et al., 2005*; *Van Der Meulen et al., 2006*, *2009*; *Volpato et al., 2008*; *Wang et al., 2012*). Furthermore, comparative studies have shown that differences in loading conditions, resulting from differences in locomotor behaviour and biomechanics, are often reflected as differences in architectural patterns between species (*Amson et al., 2017*; *Barak et al., 2013*; *Fajardo & Müller, 2001*; *Griffin et al., 2010*; *Hébert, Lebrun & Marivaux, 2012*; *MacLatchy & Müller, 2002*; *Maga et al., 2006*; *Matarazzo, 2015*; *Ryan & Ketcham, 2002*, *2005*; *Ryan & Shaw, 2012*; *Su, Wallace & Nakatsukasa, 2013*; *Tsegai et al., 2013*). Locomotor-dependent architectural differences have also been borne out in Part I of this series, which has highlighted a number of important differences in cancellous bone architecture between the hindlimb bones of humans and birds, the two kinds of obligate, striding bipeds alive today (*Bishop et al., 2018b*).

As outlined in Part I of this series, the overarching paradigm that relates cancellous bone architectural fabric to its mechanical environment is the 'trajectorial theory'. First enounced by *Wolff (1892)*, in its modern formulation the trajectorial theory states that the principal material directions of a given volume of cancellous bone are aligned with the principal stress trajectories generated from physiological loading, but only at spatial scales at which cancellous bone can be treated as a continuous material (*Cowin, 2001*). The principal material directions describe the directions in which a volume of cancellous bone is most and least stiff, whereas (continuum-level) principal stress trajectories describe how compressive and tensile forces are distributed throughout a material under a particular loading regime. As also reviewed in Part I, the principal material directions of a given volume of cancellous bone are closely aligned with its principal fabric directions, that is, the directions of strongest and weakest alignment of trabeculae (*Kabel et al., 1999*; *Odgaard et al., 1997*; *Turner et al., 1990*; *Ulrich et al., 1999*). This effectively means that the architectural fabric of cancellous bone parallels the principal stress trajectories during the normal use of a bone. Such a correspondence has been demonstrated to occur in a wide variety of instances, by both experimental (*Biewener et al., 1996*; *Lanyon, 1974*; *Su et al., 1999*) and theoretical (*Beaupré, Orr & Carter, 1990*; *Carter, Orr & Fyhrie, 1989*; *Currey, 2002*; *Gefen & Seliktar, 2004*; *Giddings et al., 2000*; *Hayes & Snyder, 1981*; *Jacobs, 2000*; *Jacobs et al., 1997*; *Koch, 1917*; *Miller, Fuchs & Arcan, 2002*; *Pauwels, 1980*; *Rudman, Aspden & Meakin, 2006*; *Sverdlova, 2011*; *Vander Sloten & Van Der Perre, 1989*) studies of locomotion.

In the aforementioned theoretical studies, the general approach was the same. That is, given a continuum-level model of the bone, apply a loading regime that reflects *in vivo*

physiological conditions (often derived from empirical measurements), calculate the resulting principal stress trajectories and then compare them to observations of cancellous bone architecture. It stands to reason that, if the trajectorial theory is true, the approach will also hold in reverse. Here, it is hypothesized that if one constructs a continuum-level model of a whole bone and seeks to determine the loading regime(s) in which principal stresses align with observed cancellous bone architecture, the resulting loading regime(s) should be physiologically realistic. It is also hypothesized that this 'reverse trajectorial approach', when framed in the context of a whole musculoskeletal system (such as a limb), should result in a physiologically realistic posture used during normal activity. If these predictions hold true, then this has the potential to provide new insight into understanding posture and locomotor biomechanics in extinct species, such as non-avian theropod dinosaurs, a group for which much interest surrounds their manner of locomotion (*Hutchinson & Allen, 2009*).

The present study aimed to test the above hypotheses, and thereby investigate the validity of the reverse approach. It focused on an extant theropod species, the chicken (*Gallus gallus*), as a generalized representative for all extant, ground-dwelling birds, for which much knowledge about terrestrial locomotor biomechanics exists. By integrating musculoskeletal and finite element modelling with observations of cancellous bone architecture, this study asked the question: 'in what posture of the hindlimb do principal stresses align with observed cancellous bone architecture, and is this posture consistent with empirical observations?' In testing the reverse approach with a modern theropod and assessing its validity, the approach may then be applied to extinct, non-avian theropods, as will be done in Part III (*Bishop et al., 2018a*). Additionally, the results of the present study can also demonstrate how applicable this approach may be for understanding locomotor biomechanics in extinct tetrapod vertebrates in general.

## MATERIALS AND METHODS

### The overall approach

The concept of using cancellous bone architectural patterns to derive *in vivo* loading regimes is not new. However, previously developed approaches (*Bona, Martin & Fischer, 2006*; *Campoli, Weinans & Zadpoor, 2012*; *Christen et al., 2012*, *2013a*, *2013b*, *2015*; *Fischer, Jacobs & Carter, 1995*; *Zadpoor, Campoli & Weinans, 2013*) are so different from that of the present study, or indeed are likely not applicable to extinct species, that an examination of these approaches will be left to the Discussion. In the present study, the approach of identifying the loading regimes and hindlimb locomotor biomechanics that reflected observed cancellous bone architecture was a repetitive one, which may be summarized as follows. For a given test posture, the forces and moments involved were first calculated using a musculoskeletal model, assuming a quasi-static situation, which were then transferred to a set of finite element models to calculate principal stress trajectories in the femur, tibiotarsus and fibula. These stress trajectories were then compared to the observed cancellous bone fabric in each bone, as reported in Part I of this series. The amount of correspondence between stress trajectories and cancellous bone fabric, and where this occurred, was then used to guide the set-up of a new test posture.

Starting from a general avian mid-stance posture, the process was repeated until new posture attempts did not produce improvement in overall correspondence compared to the previous best posture (they either produced similar or poorer levels of correspondence); at this point it was judged that no further significant improvement was likely to be gained. This repetitive procedure involved a combination of both manual and semi-automated techniques (detailed below), and in total took an estimated 350 h to perform.

The above procedure resulted in a single 'solution posture' that best reflected as much of the observed cancellous bone architecture as possible, across all three bones. In seeking a single posture, this study therefore sought the posture that engendered the greatest amount of remodelling stimulus in cancellous bone, to which the bones responded and adapted their architecture. Since bone remodelling is more responsive to repetitive, dynamic loading that produces greater peak strains, as well as higher strain rates (*Lanyon, 1996*; *Turner, 1998*), the movements in dynamic locomotion will presumably exert a strong influence on cancellous bone architecture in limb bones. It was assumed here that the loading regime during mid-stance in normal locomotion would be important for the determination of the observed cancellous bone architecture. This is because the magnitude of the ground reaction force (GRF) is substantial at around mid-stance in a wide range of animals, even if this is not when the absolute highest forces are experienced across the stance phase (*Alexander, 1977*; *Andrada et al., 2013*; *Bishop et al., 2018*; *Blob & Biewener, 2001*; *Bryant et al., 1987*; *Butcher & Blob, 2008*; *Gosnell et al., 2011*; *Hutchinson, 2004*; *Ren et al., 2010*; *Rubenson et al., 2011*; *Sheffield & Blob, 2011*; *Witte, Knill & Wilson, 2004*). Measured *in vivo* joint reaction forces are also high at around this point in the stance (*Bergmann et al., 2001*; *Bergmann, Graichen & Rohlmann, 1999*; *Page et al., 1993*; *Taylor & Walker, 2001*), as are the reaction forces when calculated using biomechanical models (*Giarmatzis et al., 2015*; *Goetz et al., 2008*; *Lerner et al., 2015*; *Modenese & Phillips, 2012*). Hence, a general avian mid-stance posture was used as an initial starting point in the modelling process; this posture was not based on any one species, but rather represented a qualitative 'average' of avian postures that have been reported in the literature.

Several simplifications or assumptions were necessary throughout the modelling and simulation process. These could have been avoided or refined if only extant theropods were the ultimate focus of the study. However, as the approach outlined here needed to be applicable to extinct, non-avian theropods as well, any limitations inherent to non-avian theropods, such as absence of data concerning soft tissues (i.e. muscles, tendons, ligaments, cartilage, menisci) also had to be observed in the chicken models and simulations. Thus, when there is good evidence of a feature or constraint in both the extinct and extant species, the attempt has been made to be a specific as possible; however, when faced with considerable uncertainty or ambiguity, a more relaxed, generalized approach was taken. Not only does this tend to invoke fewer assumptions (i.e. model simplicity), but it also enables greater consistency across species for the sake of comparison.

All scripts, models and data used are held in the Geosciences Collection of the Queensland Museum, and are available upon request to the Collections Manager.

## Skeletal geometry acquisition

This study focused on a single extant theropod species, the chicken (*G. gallus*), to explore the validity of the reverse approach. Chickens are a good generalized representative of extant, ground-dwelling birds, and much knowledge exists about their terrestrial locomotor biomechanics (*Carrano & Biewener, 1999*; *Grossi et al., 2014*; *Muir, Gosline & Steeves, 1996*; *Rose et al., 2016*). Furthermore, as demonstrated in Part I of this series (*Bishop et al., 2018b*), the cancellous bone fabric of chickens is quite typical of extant, ground-dwelling birds; in quantitative comparisons, chickens fall well within the region of space occupied by birds on streoplots of fabric directions. Thus, given the logistical constraints of time and resources, the chicken was deemed a good choice of species upon which to base the present study.

The models developed here were based on a 1.56 kg adult female chicken (white leghorn breed), which was studied previously by *Bishop et al. (2018)*. This specimen was different to the two specimens that were investigated in Part I, on account of logistical reasons. The intact carcass was subject to X-ray computed tomographic (CT) scanning (Siemens Somatom Definition AS+, 120 kV peak tube voltage, 255 mA tube current, 1,000 ms exposure time, 0.367 mm pixel resolution, 0.2 mm slice thickness), and the resulting scans were segmented in Mimics 17.0 (Materialize NV, Leuven, Belgium) via a combination of manual and automatic techniques. This produced an initial surface mesh for each bone, which was smoothed in 3-matic 9.0 (Materialize NV, Leuven, Belgium), and then refined to produce a more isoparametric mesh in ReMESH 2.1 (*Attene & Falcidieno, 2006*; http://remesh.sourceforge.net/). An isoparametric mesh is one in which the comprising triangles are all approximately equilateral in shape, and all of similar size. This is important for the generation of a volume mesh for use in finite element analyses, because the quality of the volume mesh is dependent on the quality of the surface mesh from which it is derived (*Wroe et al., 2007*).

Refined surface meshes were produced for the femur, tibiotarsus, fibula and tarsometatarsus (including metatarsal I), as well as the pelvis, sacrum and caudal vertebrae. These meshes were used in the creation of the musculoskeletal model and their derived volume meshes were used in the finite element model, facilitating complete node-to-node correspondence between the two modelling environments. Despite the patella and tarsal sesamoid being present in the chicken, they were not included in the development of the models, both for the sake of simplicity and also to maintain consistency with models developed for non-avian theropods (in Part III), which lack these bones. They did, however, help inform the construction of lines of action of muscles that crossed the knee and ankle joints in the musculoskeletal model. In light of recent advances in understanding of patellar mechanics in extant birds (*Allen et al., 2017*; *Regnault et al., 2017*), future studies may be able to take the patella into account, although this would reduce comparability between models of extant birds and non-avian theropods, the latter of which lacked patellae.

## Musculoskeletal model development

A musculoskeletal model of the right hindlimb of the chicken was constructed in NMSBuilder (*Martelli et al., 2011*; *Valente et al., 2014*) for use in OpenSim 3.0.1

(*Delp et al., 2007*), and is shown in Fig. 1. It comprised 12 degrees of freedom and 38 musculotendon actuators.

### Definition of joints

The pelvis, sacrum and caudal vertebrae were fixed relative to each other and relative to the global reference frame, forming a single 'pelvis' segment. They were oriented such that a line through the neural canal of the anterior sacral vertebrae was horizontal and the postacetabular pelvis sloped ventrally, comparable to the orientation of the pelvis of ground-dwelling birds during stance and gait (*Andrada et al., 2013*; *Gatesy, 1999a*; *Rubenson et al., 2007*). Although the orientation of a bird's pelvis can vary during the stride and across different speeds of locomotion (*Abourachid et al., 2011*; *Gatesy, 1999a*; *Rubenson et al., 2007*), as a modelling simplification the position or orientation of the pelvis segment, defined by six of the 12 model degrees of freedom (three translational, three rotational), was fixed in all simulations.

The hip joint was modelled as a ball-and-socket joint with three degrees of freedom, namely flexion–extension, adduction–abduction and long-axis rotation. The three axes of rotation were initially parallel to the axes of the global coordinate system (+$x$ is anterior, +$y$ is medial, +$z$ is dorsal), and the order of rotation was flexion–extension, followed by adduction–abduction, followed by long-axis rotation. The centre of the joint in the femur was determined by fitting a sphere to the femoral head in 3-matic, and the centre of the joint in the acetabulum was determined by fitting a sphere to the concave articular surface in 3-matic. The femur was then positioned relative to the pelvis such that the joint centres of the femur and acetabulum were coincident. The 'neutral' orientation of the femur with respect to the pelvis (i.e. where all hip joint angles are zero) was such that the standard anatomical directions for the bone were set parallel to the axes of the global coordinate system (+$x$ is anterior, +$y$ is medial, +$z$ is proximal). The neutral orientations for all bones distal to the femur were set by how they articulated with their neighbouring proximal bone.

For simplicity, the knee joint was modelled with a single degree of freedom representing flexion–extension, although it is acknowledged that in reality the avian knee is also capable of significant abduction-adduction and long-axis rotation movement (*Kambic, Roberts & Gatesy, 2014*; *Rubenson et al., 2007*). No translation of the flexion–extension axis was permitted (i.e. it was fixed relative to the femur), neither was any relative movement between the tibiotarsus and fibula. The orientation and position of the flexion–extension axis relative to the femur, tibiotarsus and fibula, and the orientation and position of the tibiotarsus and fibula relative to the femur, was determined manually. This ensured that there was realistic alignment and movement of the bones across the physiological range of flexion–extension. For example, the tibiofibular crest of the lateral femoral condyle followed the space between the tibiotarsus and fibula at high flexion angles; no bone interpenetration occurred at any orientation; and the amount of space between the tibiotarsus and femur, and between the fibula and femur, remained fairly constant across the range of motion (i.e. conservation of volume of the intervening soft tissues). Additionally, the alignment of the bones was compared to their in situ orientations in the left and right limbs of the intact carcass, as observed from the CT scans. Asymmetry in the size and shape of the distal

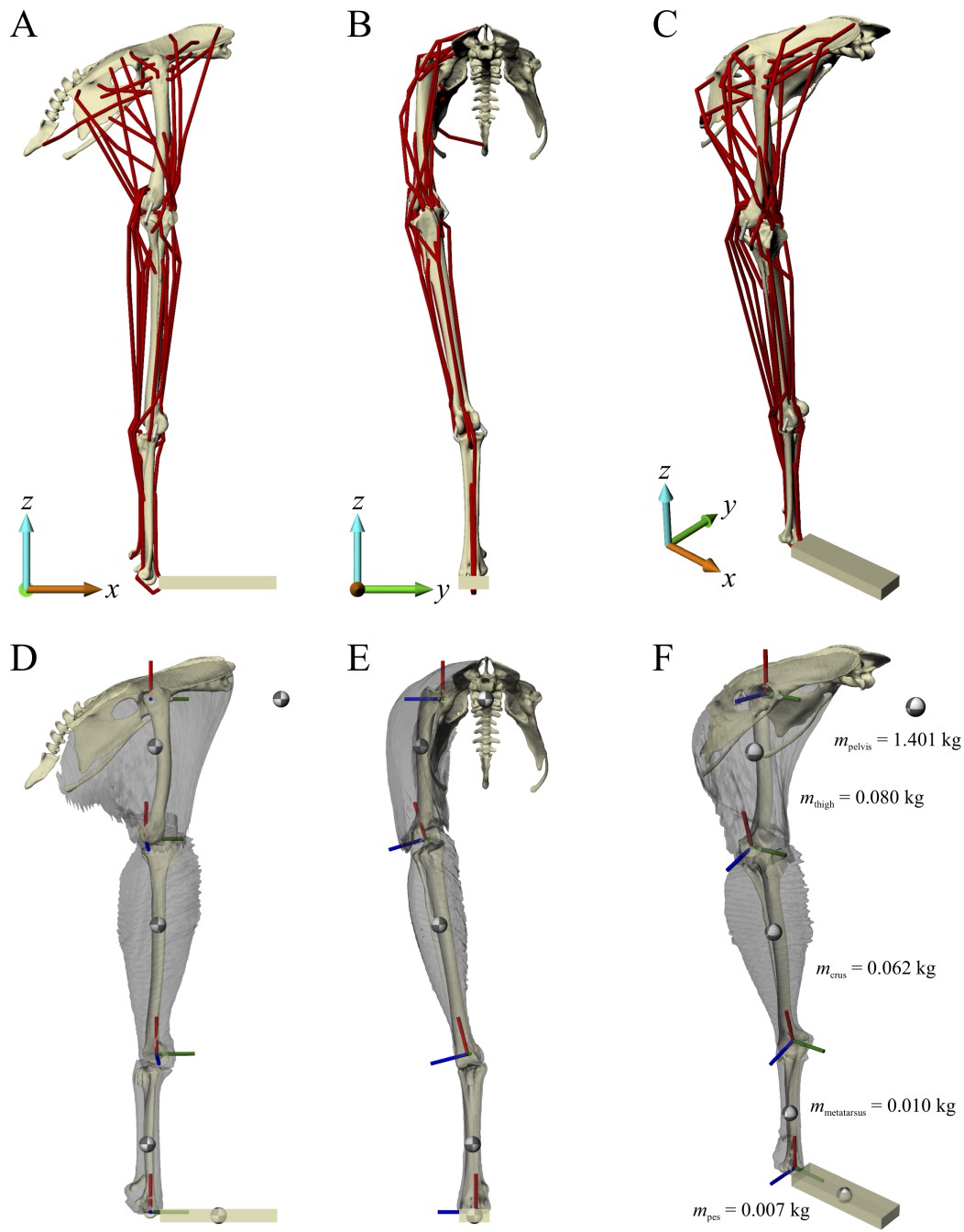

**Figure 1 The musculoskeletal model of the chicken hindlimb developed in this study.** This is shown in the 'neutral posture' for all joints, that is, when all joint angles are zero. (A–C) Geometries of the musculotendon actuators in relation to the bones, in lateral (A), anterior (B) and oblique anterolateral (C) views. (D–E) Location and orientation of joint coordinate systems (red, green and blue axes), the centres of mass for each segment (grey and white balls) and the soft tissue volumes, derived from CT scans and used to calculate mass properties; these are shown in the same views as (A–C). Also reported in (F) are the masses for each segment; the pelvis segment represents the body as well as the contralateral limb. In (D–F), the flexion–extension axis of each joint is the blue axis. For scale, the length of each arrow in the triad of the global coordinate system is 40 mm.

femoral condyles inherently meant that when the femur was in the neutral orientation, the knee joint axis was angled slightly mediolaterally in the coronal plane (Fig. 1E). Consequently, this also meant that in the neutral orientation, the distal end of the tibiotarsus and fibula were angled in towards the body midline (Figs. 1B and 1E).

Given the likely sizeable quantities of cartilage and menisci in the knee joint of extinct, non-avian theropods (Bonnan et al., 2010, 2013), and the fact that the present study needed to be wholly consistent with any modelling limitations inherent to non-avian theropods, it was felt that this representation of the knee would be more reliable than a strictly objective, geometry-based definition of the joint axis (Brassey, Maidment & Barrett, 2017; Hutchinson et al., 2005, 2008; although these studies did include space for soft tissues). This is because such a definition is only based on the available bony geometry, which may not fully reflect the actual range of possible joint movement. Moreover, such a definition only uses half of the contributing joint surfaces, for example using the femur whilst ignoring the tibiotarsus and fibula. Differences in how the knee joint is defined would be expected to have an influence on both the orientation of the knee joint axis relative to the bones and the neutral orientation of the limb.

The ankle and metatarsophalangeal joints were both modelled with a single flexion–extension degree of freedom, although as for the knee it is acknowledged that this is a simplification of reality (Kambic, Roberts & Gatesy, 2014). As for the knee joint, no translation of the flexion–extension axis was permitted in either joint; the ankle axis was fixed relative to the tibiotarsus, and the metatarsophalangeal axis was fixed relative to the tarsometatarsus. The flexion–extension axis of the ankle joint was determined in 3-matic by fitting a cylinder to the outer margins of the articular surfaces of the tibiotarsus, with the axis of the cylinder taken to be the axis of movement. The flexion–extension axis of the metatarsophalangeal joint was taken to be parallel to the $y$-axis when the limb was in a neutral orientation. Care was taken to ensure that bone interpenetration did not occur at these joints as well, over the range of joint motion typically reported for avian terrestrial locomotion in the literature. Metatarsal I was fixed relative to the tarsometatarsus, and digit I was not modelled.

The pes (digits II–IV) was modelled as a rectangular prism, parallel to the axes of the global reference frame in the neutral limb orientation, as done by Hutchinson et al. (2005, 2008). This was not only for model simplicity, but also because of the uncertainty surrounding the topology and degree of differentiation of pedal muscles in non-avian theropods (Carrano & Hutchinson, 2002; Hutchinson, 2002). Hence, for consistency, these modelling limitations inherent to non-avian theropods were also observed in the chicken model. The length of the prism was set as the total length of digit III, and the width set as the mediolateral width of the distal tarsometatarsus, across the condyles.

### Definition of muscle and ligament anatomy

A total of 34 musculotendon actuators were used to represent muscles in the model; an additional four actuators were used to represent the medial and lateral collateral ligaments of the knee and ankle, thus allowing the possibility of 'passive' forces to be included. The origins and insertions of the actuators in the model (Table 1) were derived

**Table 1 The origins and insertions of each of the muscles and ligaments represented in the chicken musculoskeletal model.**

| Muscle or ligament | Abbreviation | Origin | Insertion |
|---|---|---|---|
| Iliotibialis cranialis | IC | Anterior rim of dorsal iliac crest | Patellar tendon [medial aspect of anterior cnemial crest] |
| Iliotibialis lateralis pars preacetabularis | ILPR | Dorsolateral iliac crest, anterior to acetabulum | Patellar tendon [anterior cnemial crest] |
| Iliotibialis lateralis pars postacetabularis | ILPO | Dorsolateral iliac crest, posterior to acetabulum | Patellar tendon [anterior cnemial crest] |
| Ambiens | AMB | Preacetabular process on proximal pubis | Lateral fibular head |
| Femorotibialis externus | FMTE | Lateral femoral shaft | Patellar tendon [anterior cnemial crest] |
| Femorotibialis medius | FMTM | Anterior femoral shaft | Patellar tendon [anterior cnemial crest] |
| Femorotibialis internus | FMTI | Medial femoral shaft | Patellar tendon [medial aspect of anterior cnemial crest] |
| Iliofibularis | ILFB | Lateral postacetabular ilium, anterior to FCLP | Fibular tubercle |
| Flexor cruris lateralis pars pelvica | FCLP | Lateral surface of posterior end of ilium and adjacent caudal vertebrae | Medial proximal tibiotarsus |
| Flexor cruris lateralis pars accessoria | FCLA | From FCLP | Distal posterior femur |
| Flexor cruris medialis | FCM | Lateral surface of posterior end of ischium | Medial proximal tibiotarsus |
| Iliofemoralis externus | IFE | Processus supratrochantericus of ilium | Trochanteric shelf of femur |
| Iliofemoralis internus | IFI | Ventral preacetabular ilium, ventral to ITM | Medial surface of proximal femur (distal to femoral head) |
| Iliotrochantericus cranialis | ITCR | Ventral preacetabular ilium | Anterolateral surface of femoral trochanter, distal to ITC |
| Iliotrochantericus medius | ITM | Ventral preacetabular ilium, posterior to ITCR | Anterolateral surface of femoral trochanter, distal to ITC |
| Iliotrochantericus caudalis | ITC | Lateral surface of preacetabular ilium | Anterolateral surface of femoral trochanter |
| Ischiofemoralis | ISF | Lateral ischium | Lateral proximal femur |
| Caudofemoralis pars caudalis | CFC | Ventrolateral surface of pygostyle | Posterior surface of proximal femoral shaft |
| Caudofemoralis pars pelvica | CFP | Lateral ilium, posterior to ILFB and dorsal to ISF | Posterior surface of proximal femoral shaft (lateral to CFC) |
| Obturatorius medialis | OM | Medial surfaces of ischium and pubis | Posterolateral surface of proximal femur |
| Puboischiofemoralis pars lateralis | PIFL | Ventral ischium and pubis | Posterior surface of femoral shaft, lateral to PIFM |
| Puboischiofemoralis pars medialis | PIFM | Ventral ischium and pubis (ventral to PIFL) | Posterior surface of femoral shaft, medial to PIFL |
| Gastrocnemius lateralis | GL | Lateral aspect of distal femur (proximal to lateral condyle) | Posterior surface of tarsometatarsus |
| Gastrocnemius intermedia | GI | Medial aspect of distal femur (near medial condyle) | Posterior surface of tarsometatarsus |
| Gastrocnemius medialis | GM | Anteromedial proximal tibiotarsus | Posterior surface of tarsometatarsus |
| Flexor digitorum longus | FDL | Caudal surface of tibiotarsus | Ventral aspect of digit II-IV phalanges |
| Other digital flexors* | ODF | Caudal femur, near lateral condyle (but proximal to it) | Ventral aspect of digit II-IV phalanges |
| Flexor hallucis longus | FHL | Caudal distal femur, popliteal fossa region | Ventral aspect of phalanx II-2 (ungual) |

(Continued)

| Muscle or ligament | Abbreviation | Origin | Insertion |
|---|---|---|---|
| Extensor digitorum longus | EDL | Anterior surface of tibiotarsus, distal to TCT origin | Dorsal aspect of digit II-IV phalanges; passes under pons supratendinous |
| Other digital extensors** | ODE | Anterior aspect of tarsometatarsus | Dorsal aspect of digit II-IV phalanges |
| Tibialis cranialis caput femorale | TCF | Distal lateral condyle of femur | Anterior proximal tarsometatarsus |
| Tibialis cranialis caput tibiale | TCT | Distal aspect of anterior cnemial crest | Anterior proximal tarsometatarsus |
| Fibularis longus | FL | Soft tissues surounding proximolateral tibiotarsus [apex of lateral cnemial crest] | Tendon of flexor perforati digiti III [modelled separately, to insert on ventral pes] |
| Fibularis brevis | FB | Anterolateral tibiotarsus and anteromedial fibula | Lateral proximal tarsometatarsus |
| Knee medial collateral ligament | KMCL | Depression on medial surface of medial femoral condyle | Medial proximal tibiotarsus, proximal to FCLP and FCM insertions |
| Knee lateral collateral ligament | KLCL | Depression on lateral surface of lateral femoral condyle | Lateral fibular head, proximal to AMB insertion |
| Ankle medial collateral ligament | AMCL | Depression on medial surface of medial condyle of tibiotarsus | Medial proximal tarsometatarsus |
| Ankle lateral collateral ligament | ALCL | Depression on lateral surface of lateral condyle of tibiotarsus | Lateral proximal tarsometatarsus, anterior to FB insertion |

Notes:
Those muscles that attach to the patella or patellar tendon were modelled as attaching in a general fashion to the apices of one of the cnemial crests (identified in brackets).
* ODF includes the flexores perforantes et perforatus digitorum II et III and flexores perforatus digitorum II, III et IV.
** ODE includes the extensores brevis digitorum III et IV and extensor proprius digiti III.

from first-hand observations made during dissections (from four individuals in total), as well as comparison to the published literature (*Hudson, 1937*; *Hudson, Lanzillotti & Edwards, 1959*; *Paxton et al., 2010*), and were placed as near as possible to the centroid of the area of attachment in each case. The 3D course of each actuator from origin to insertion was constrained to follow anatomically realistic paths as observed during dissections and reported in the literature. This was achieved through the placement of a number of intermediate 'via points' (*Delp et al., 1990*) along the course of the actuator. Only the minimum number of via points was used to achieve realistic paths, across the whole physiological range of motion.

For the purposes of the current study, a number of simplifications were made regarding the representation of some of the muscles:

1. The popliteus was not included in the model, for it runs between the proximal tibiotarsus and fibula. Since relative movement between the two bones was not modelled here, inclusion of the popliteus is unnecessary.

2. The plantaris was not included in the model, because it runs from the proximal tibiotarsus to the medial aspect of the tibial cartilage surrounding the ankle; it was therefore considered unlikely to play a significant role in load bearing, and thus load transmission to the bones. For a similar reason, the secondary attachment of the fibularis longus (FL) to the tibial cartilage was also not modelled.

3. On account of its small size, similar line of action to, and common insertion with, the obturatorius medialis (OM), the obturatorius lateralis was not modelled: a single musculotendon actuator was deemed sufficient to represent the two muscles.

4. Both parts of the flexor cruris lateralis (pars pelvica, FCLP; pars accessoria, FCLA) were modelled with separate musculotendon actuators. At the point where the FCLA diverges from the FCLP, the actuators went their own separate way towards their respective insertions, but proximal to this, they took on the same line of action towards the origin on the pelvis. A similar approach was used for modelling the two heads of the tibialis cranialis (caput femorale; caput tibiale).

5. The flexor hallucis brevis and extensor hallucis longus were not modelled, because they run from the tarsometatarsus to the ungual of digit I; as there was no degree of freedom that these two muscles could influence in the model, they were unnecessary.

6. As noted above, there is considerable uncertainty surrounding the topology and degree of differentiation of many of the digital flexor and extensor muscles in non-avian theropods (*Carrano & Hutchinson, 2002*; *Hutchinson, 2002*). The representation of these muscles in the chicken model was consequently simplified, to maintain consistency with non-avian theropod models, but also because the pes was modelled as a single unit. The flexor digitorum longus (FDL) and flexor hallucis longus were modelled separately, but the deep digital flexors were represented by a single musculotendon actuator 'other digital flexors' (ODF), which grossly reflected the lines of action of the individual muscles. Likewise for the extensors, the extensor digitorum longus was modelled separately, but the deep digital extensors were represented by a single musculotendon actuator other digital extensors (ODE).

7. Owing to the simplified representation of the deep digital flexors, the main insertion of the FL was extended to the ventral aspect of the pes segment.

These modelling simplifications were not expected to have any significant influence on the loading conditions experienced by the femur, tibiotarsus or fibula.

The 38 musculotendon actuators so modelled here provided the forces necessary to counter collapse of the hindlimb during the simulation of a given test posture. Whilst the maximum force able to be produced by each muscle (or resisted by each ligament) could be estimated from empirical anatomical data (*Calow & Alexander, 1973*; *Hutchinson et al., 2015*; *Lamas, Main & Hutchinson, 2014*), this is obviously not possible in the case of extinct, non-avian theropods. As such, for the sake of simplicity and consistency across extinct and extant species, all musculotendon actuators were assigned the same maximum force, 30.597 N, equal to two times body weight (BW). A value of 2 BW was chosen because some muscles would have undoubtedly been capable of exerting forces of that magnitude, or greater, as is the case in other animals (*Anderson & Pandy, 1999*; *Charles et al., 2016*; *Hutchinson et al., 2015*; *O'Neill et al., 2013*; *Smith et al., 2006*). With all actuators having the capacity to exert forces of that magnitude, there was ample force for the actuation of each degree of freedom, obviating the need for reserve actuators (but see section 'Simulation and calculation of internal forces and moments' below).

*Definition of segment mass properties*

As a means to estimating the mass properties of each limb segment in the musculoskeletal model, the flesh surrounding each limb bone was segmented from the carcass CT scans in Mimics to produce a series of surface meshes. Using the computer-aided design software Rhinoceros 4.0 (McNeel, Seattle, WA, USA), each flesh mesh was then repositioned in space to align it with the underlying bone(s) in their neutral orientation. Additionally, the thigh segment flesh was retro-deformed to fit the pelvis and femur in the neutral pose, and care was taken to ensure that the net change in volume was negligible; this process was accomplished in Rhinoceros using the 'cage edit tool', a form of host mesh warping (*Fernandez et al., 2004*). The mass and centre of mass (COM) of each segment was then able to be calculated in NMSbuilder, assuming a bulk density of 1,000 kg/m$^3$. The total mass of the right hindlimb in the model was 0.159 kg, and therefore the mass of the remaining body was 1.401 kg; this was designated as the mass of the pelvis segment in the model. Given the data reported by *Allen et al. (2013)*, the combined COM of the whole body, minus the right leg, in their geometric model of a chicken was 0.076 m anterior to the hip joint. Scaling isometrically (via femur length) to the chicken specimen modelled here, the COM is 0.068 m anterior to the hip; this was taken to be the location of the COM of the pelvis segment in the musculoskeletal model. Since the orientation of the pelvis segment was fixed in all simulations, and all simulations were quasi-static, the only moment produced by the pelvis segment would be that by virtue of its weight, and consequently the dorsoventral position of the pelvis segment COM would not matter. As such, the dorsoventral position of the COM of the pelvis segment was assumed to be level with the hip. Moments of inertia for each segment were not calculated, on account that the simulations performed in this study were quasi-static only.

## Musculoskeletal simulations

### Deriving a test posture

Based on the argument presented in section 'The overall approach' above, a general mid-stance posture was used as an initial starting posture, which was then modified in subsequent modelling attempts. It was based on comparison to the kinematic data previously reported for ground-dwelling birds (*Abourachid & Renous, 2000*; *Gatesy, 1999a*; *Grossi et al., 2014*; *Reilly, 2000*; *Rubenson et al., 2007*; *Stoessel & Fischer, 2012*): hip extension of −30° below horizontal, hip abducted 5° from midline, hip rotated 20° externally, knee flexed 93° from neutral position, ankle flexed 46° from neutral position, metatarsophalangeal joint extended 16° from neutral position. The modification of a given test posture to produce a new posture at the start of a new modelling attempt followed hierarchical priorities: hip extension angle > knee angle > ankle and metatarsophalangeal angles, with the metatarsophalangeal angle set so as to position the pes segment flat on the ground (i.e. parallel to the *x*–*y* plane). Each posture was also constrained by thee basic criteria:

1. No interpenetration occurred between any bones, including those of the pelvis.
2. The centroid of the pes segment, taken to be the location of the centre of pressure (COP) of the GRF (see below), was underneath the whole-body COM in the *x*–*z* plane.

This constraint predominantly affected the knee, ankle and metatarsophalangeal joint angles, and was necessitated by the fact that the applied GRF in the simulations was vertically oriented (see below).

3. The mediolateral step width, defined as twice the distance from the centroid of the pes segment to the body midline, was less than 15% of the posture's hip height, defined as the vertical distance from the hip joint centre to the base of the pes segment. This constraint predominantly affected the hip adduction–abduction and long-axis rotation angles, and was based on the results of *Bishop et al. (2017b)*.

### External forces

In the present study, a given test posture was analysed as a quasi-static system. Dynamic effects such as segment accelerations were not considered, as this requires additional information and assumptions about movement, which are currently unknown for extinct, non-avian theropods. (Furthermore, incorporating dynamic effects might not actually lead to a marked change in model results, *Anderson & Pandy, 2001*; *Rankin, Rubenson & Hutchinson, 2016*.) Hence, the only acceleration in the simulation was that due to gravity, of magnitude 1 BW. In order for static equilibrium to be maintained, and also to refrain from using residual actuators of the six degrees of freedom at the pelvis, this necessitated the applied GRF to be vertical and also of magnitude 1 BW (Fig. 2). This in turn required one of the following three scenarios to also be true:

1. The centroid of the pes segment (taken as the COP of the GRF) must be directly underneath the whole-body COM, in both the $x$ and $y$ directions. However, the whole-body COM is on (or almost on) the body midline, meaning that in such a scenario the pes is also on the body midline. This is posturally inaccurate, because theropods employ non-zero step widths across most speeds (*Bishop et al., 2017b*).

2. The centroid of the pes segment is not directly underneath the whole-body COM, instead having a non-zero step width. This is more posturally accurate, but static equilibrium will not be achieved unless:

   a) The COP is moved away from the centroid of the pes and retained directly under the whole-body COM. This is more speculative however, because empirical data on the path of the COP in modern bipeds shows that it remains close to the centre of the foot, not straying too far laterally or medially away from the foot midline (*Schaller et al., 2011*; *Winter, 2009*).

   b) The COP is kept at the centroid of the pes, and an additional moment about the $x$-axis is applied to the pes. This moment is equal to the product of BW and the mediolateral distance between the COM and COP:

   $$M_x = m \cdot g \cdot \left( COP_y - COM_y \right), \tag{1}$$

   where $m$ is body mass and $g$ is the acceleration due to gravity, 9.81 m/s$^2$. This is physiologically implausible however, as in reality the feet can only be capable of applying a moment about the vertical ($z$) axis, the so-called 'free moment'.

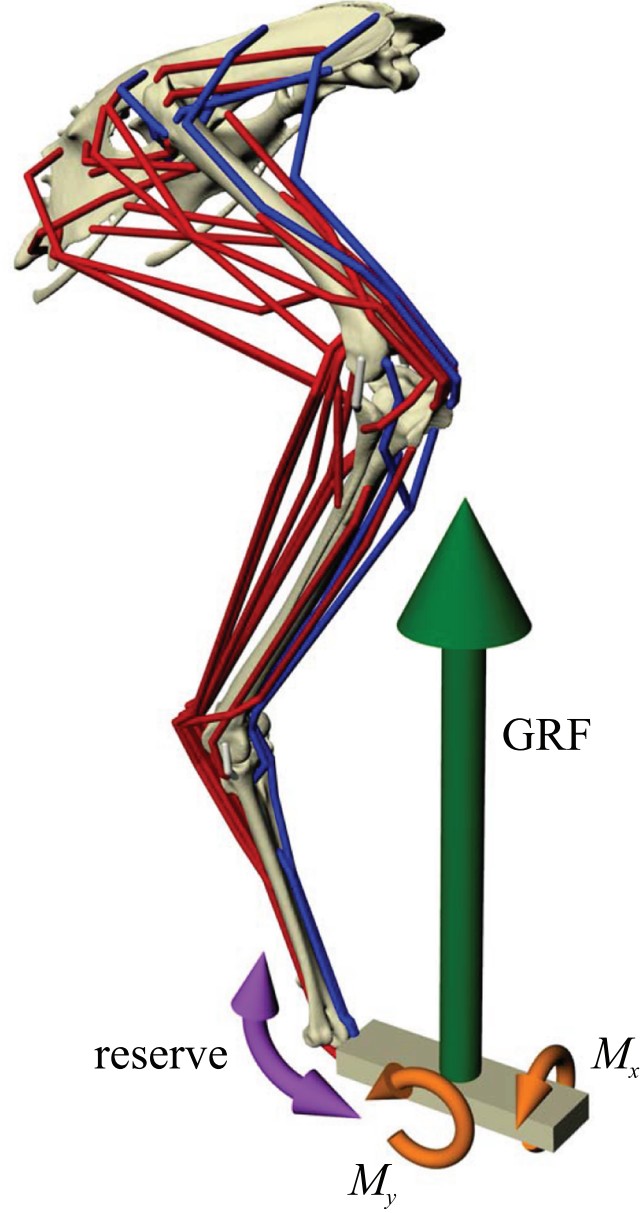

**Figure 2 Musculoskeletal simulation of a given test posture.** Muscles that are active are red, whilst those set to be inactive during simulation are blue. External loads applied to the pes segment are the vertical ground reaction force (GRF) and moments about the $x$ and $y$ axes ($M_x$ and $M_y$, respectively). A reserve actuator is also applied to the metatarsophalangeal joint (purple). Loads are not shown to scale.

Hence, regardless of which scenario is used, some amount of accuracy must be lost in order for static equilibrium to be achieved and the simulation to be solved. The present study followed scenario 2(b), in order to maintain postural accuracy (Fig. 2). An additional moment about the $y$-axis ($M_y$) was also applied to the pes to account for minute positional discrepancies between the $x$-coordinates of the COP and COM (i.e. when the COP was not exactly underneath the COM), but this never amounted to

more than 0.011 Nm (<1% of the product of BW and COM height) in any of the simulations performed.

That the applied GRF in the simulations was vertical is appropriate in the context of the current study for two reasons. Firstly, in a wide range of animals including theropods the GRF is approximately vertical, in the sagittal plane, at around mid-stance (*Bishop et al., 2018*; *Hutchinson, 2004*). Secondly, in a wide range of animals including theropods the GRF is largely vertical at the instance of peak net GRF (especially in more 'running-like' gaits), and this instance also occurs at around mid-stance (*Bishop et al., 2018*). However, when the GRF is at its most vertical in the sagittal plane, or when it is at maximum magnitude, it is almost always never 1 BW in magnitude; it is sometimes a little lower, but most often it is higher, and sometimes much higher, than 1 BW. This is not a problem for the current study, because principal stress trajectories do not reflect the absolute magnitudes of applied forces, only their relative magnitudes and directions, provided that deformation remains within the elastic range (*Beer et al., 2012*). Moreover, in having the GRF as 1 BW in magnitude, this also facilitates size-independent comparisons across postures and across species following simulation.

### Simulation and calculation of internal forces and moments

Once a test posture was established and the GRF (and associated moments) was applied, the forces developed by the musculotendon actuators to resist limb collapse were calculated in OpenSim. Although 34 muscles were represented in the musculoskeletal model, not all of them would be active and exerting force at around the mid-stance of a stride. Thus, some muscles were set to be inactive in the simulations (Fig. 2; Table 2). Which muscles were set to be inactive was determined through reference to published electromyography data for birds (*Gatesy, 1990*, *1994*, *1999b*; *Jacobson & Hollyday, 1982*; *Marsh et al., 2004*; *Roberts, Chen & Taylor, 1998*). Muscles that are active only in the swing phase, or active in the stance phase but only at the very beginning or end, were considered inactive. If no data existed for a particular muscle, the following line of reasoning was employed. If the muscle belonged to the same functional group as another muscle that had been investigated (e.g. femorotibialis externus; puboischiofemoralis lateralis), its activity was assigned based on the recorded muscle. Failing that, if the muscle was considered unlikely to be involved in limb support at around mid-stance (e.g. ankle flexors, digital extensors, OM, IFI), it was considered inactive. If its activity still remained equivocal after that, then it was included in the model and deemed to be active, to be conservative. All four collateral ligament actuators were also included, to allow for passive forces to occur. These were modelled simply as linear 'reserve' actuators without incorporation of slack length or elasticity.

On account of the unknowable properties of muscle and ligament in extinct theropods, intrinsic force-length-velocity relationships were ignored for all musculotendon actuators in the simulations. That is, the actuators simply modelled the application of a force along a line of action set by the actuator geometry, defined in section 'Definition of

**Table 2  Assumed activities of the muscle actuators used in the simulations.**

| Muscle | Activity |
|--------|----------|
| IC | O |
| ILPR | O |
| ILPO | X |
| AMB | X |
| FMTE | X |
| FMTM | X |
| FMTI | X |
| ILFB | X |
| FCLP | X |
| FCLA | X |
| FCM | X |
| IFE | O |
| IFI | O |
| ITCR | O |
| ITM | X |
| ITC | X |
| ISF | X |
| CFC | X |
| CFP | X |
| OM | O |
| PIFL | X |
| PIFM | X |
| GL | X |
| GI | X |
| GM | X |
| FDL | X |
| ODF | X |
| FHL | X |
| EDL | O |
| ODE | O |
| TCF | O |
| TCT | O |
| FL | X |
| FB | X |

Note:
  X = active (capable of exerting up to 30.597 N of force), O = inactive (exerts zero force).

muscle and ligament anatomy' above. Hence, the moment $M_i$ a given actuator exerted about a given degree of freedom $i$ was equal to

$$M_i = a \cdot F_{\mathrm{max}} \cdot r_i, \tag{2}$$

where $F_{\mathrm{max}}$ is the maximum force capable of being produced (set at 2 BW), $r_i$ is the moment arm of the actuator and $a$ is the activation of the actuator, which can vary between

0 and 1. The forces developed in each musculotendon actuator were calculated using the static optimization routine of OpenSim, which solved the statically indeterminate problem of force distribution by minimizing the sum of squared activations across the actuators (*Pedotti, Krishnan & Stark, 1978*; *Rankin, Rubenson & Hutchinson, 2016*). It was found that in no simulation did the activation of any musculotendon actuator ever approach 1; indeed, activations rarely exceeded 0.5. Coupled with the omission of intrinsic force-length-velocity relationships, this prevented non-linearities from occurring in the static optimization routine, further facilitating size-independent comparisons across postures and across species post analysis. Due to the simplified representation of the pes segment and the muscles that cross the metatarsophalangeal joint, a reserve actuator was also applied to the metatarsophalangeal joint in the static optimization, with a maximum output set at 1,000 Nm (Fig. 2). This high value provided ample control of the metatarsophalangeal joint, and helped reduce excessively high and unrealistic recruitment of the few modelled musculotendon actuators that crossed the joint (FDL, ODF and FL). Across the range of postures tested, the reserve actuator never provided a moment greater than 0.41 Nm (<12% of the product of BW and COM height). In addition to the actual calculated forces, the line of action of all musculotendon actuators was also extracted from the posture, using the MuscleForceDirection plugin for OpenSim (*van Arkel et al., 2013*). Following the calculation of muscle and ligament forces, joint forces and moments were extracted using the JointReaction tool in OpenSim (*Steele et al., 2012*). All forces were extracted and expressed in the global coordinate system.

## Finite element simulations

Two finite element simulations were performed for each test posture in ANSYS 17.0 (Ansys, Inc., Canonsburg, PA, USA), one of the femur and one of the tibiotarsus + fibula. The loads applied in these simulations were exactly the same as those calculated in the musculoskeletal simulations. Furthermore, the nodes on each bone to which muscle or ligament forces were applied in the finite element simulations were the exact same nodes to which the musculotendon actuators attached in the musculoskeletal simulations. This ensures complete correspondence between the two sets of simulations.

### *Geometry*

The relative positions and orientations of each bone in the musculoskeletal simulations were maintained exactly in the finite element simulations. In addition to the modelling the focal bone (or bones) of interest, two extrinsic structures were created to represent the adjacent articulating bones, to more realistically model the distribution of joint forces (Figs. 3A and 3B). For the femur simulation, an acetabulum structure (derived from the pelvis surface mesh) and proximal crus structure (derived from the tibiotarsus and fibula surface meshes) were created. For the tibiotarsus + fibula simulation, a distal femur structure (derived from the femur surface mesh) and proximal tarsometatarsus structure (derived from the tarsometatarsus surface mesh) were created. These structures were generated simply by trimming their parent surface meshes down to the immediate area involved in the joint articulation, using a combination of Rhinoceros and 3-matic.

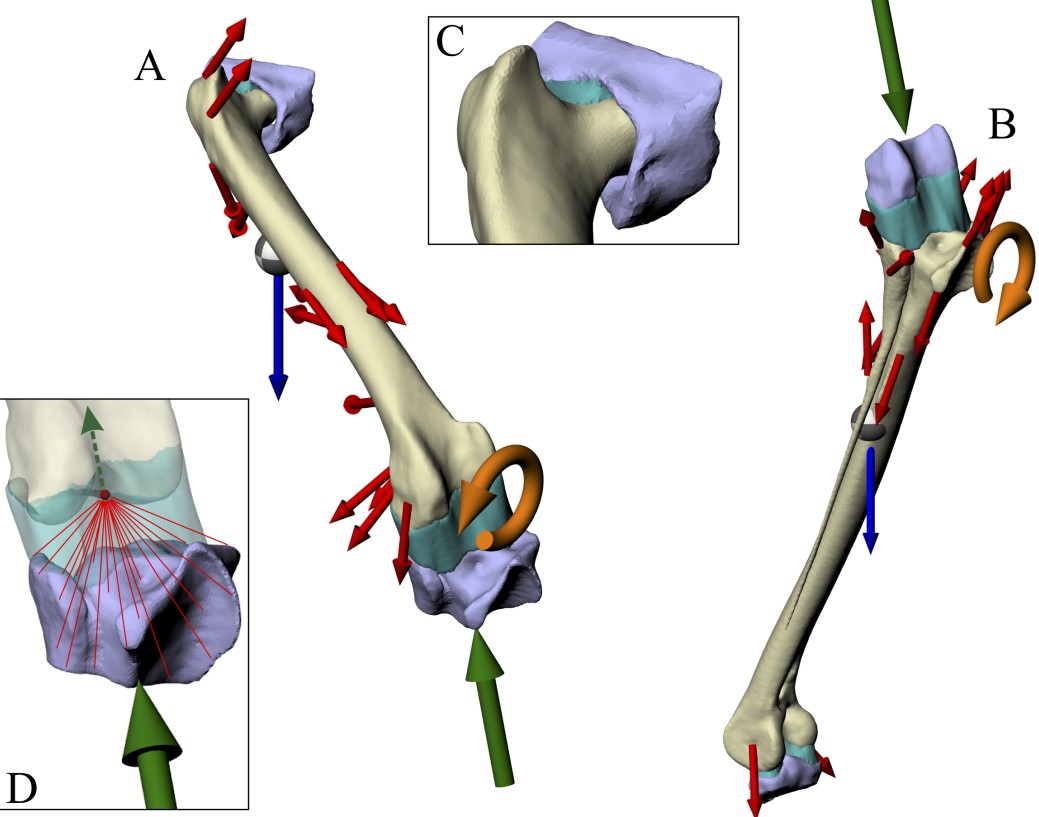

**Figure 3 Geometry, forces and constraints involved in the finite element analysis of a given test posture.** (A and B) For each posture, two simulations were performed, one for the femur (A) and one for the tibiotarsus + fibula (B). Muscle and ligament forces are red, segment weights are blue, joint forces are green and joint moments are orange. The focal bones in each simulation were 'bookended' between their adjacent articulating bones, to which restraints or joint forces were applied. (C) The intervening soft tissues between focal bones and their neighbouring bones were modelled as a single homogenous volume (turquoise). (D) Knee joint forces were applied as a remote force: the force was applied to a remote point (knee joint centre, red dot), which was topologically attached to a neighbouring bone via constraint equations (red lines, schematic illustration only). Loads are not shown to scale.

Additionally, in the proximal crus structure, the geometry was modified distally, well away from the articular areas, to fuse the tibiotarsus and fibula together, limiting movement between the two during the simulations.

In order to model the distribution of joint forces more realistically and evenly across opposing joint surfaces, the intervening soft tissues that occur between a focal bone and its neighbouring bone in life were modelled as a single volume (Fig. 3). A single, homogenous entity was chosen to represent these joint soft tissues (e.g. cartilage, menisci) in the current study, as the anatomy of such tissues is unknown for extinct, non-avian theropods. Moreover, this modelling simplification makes the analyses more tractable for the current study, instead of involving more complex, non-linear behaviours and contact formulations. The volume of soft tissues for each of the hip, knee and ankle joints was produced by connecting up the closest parts of the articular surfaces of the bones involved, using the 'loft' tool in Rhinoceros to create an initial mesh, which was then

smoothed and remeshed in 3-matic. In addition to more realistically modelling joint load distribution, the approach used here also allowed for boundary conditions (restraints) to be moved away from the bone (or bones) of interest, reducing the incidence of artifacts in the model results (Saint-Venant's principle: *Dumont, Piccirillo & Grosse, 2005*; *Gilbert, Snively & Cotton, 2016*; *McHenry et al., 2007*). It is conceptually similar to the approach employed by Phillips and co-workers in their finite element modelling of human limb bones (*Geraldes, Modenese & Phillips, 2016*; *Geraldes & Phillips, 2014*; *Phillips, Villette & Modenese, 2015*), although the actual formulations involved are markedly different.

Volume meshes for finite element analysis were generated from the surface meshes of each bone and soft tissue entity in 3-matic. All volume meshes were composed exclusively of low-order (4-node) tetrahedral elements. Meshes composed of high-order (10-node) elements may produce more accurate results than those composed of low-order elements, but this discrepancy decreases with a greater number of elements used (*Bright & Rayfield, 2011*; *Dumont, Piccirillo & Grosse, 2005*). Furthermore, considering the relatively simple geometry of the bones being modelled here, any such discrepancy was considered to be minimal. In producing the volume meshes, the maximum tetrahedral edge length was constrained, so as to avoid the generation of tetrahedral elements of undesirably high aspect ratios, which can lead to inaccurate results. The maximum edge length for each entity was defined as being no more than double the mean edge length of the triangles in the parent surface mesh. The mean edge length of the surface mesh triangles was calculated as

$$L = \sqrt{\frac{4A}{\sqrt{3}n}}, \tag{3}$$

where $A$ is the total area of the surface mesh and $n$ is the number of comprising triangles in the surface mesh. This assumes that the average triangle in the surface mesh is equilateral in shape[1]. The total number of elements used across the various postures tested ranged from 803,508 to 822,322 in the femur simulation and from 986,280 to 1,005,550 in the tibiotarsus + fibula simulation. Although a convergence analysis was not conducted, it was considered that this was a sufficient number of elements for the current study, given the relatively simple geometry of the structures being modelled (*Bright & Rayfield, 2011*; *Gilbert, Snively & Cotton, 2016*).

In the finite element simulations, the interfaces of adjacent contacting entities (e.g. hip soft tissues and femur) were fixed relative to each other using a bonded contact formulation ANSYS, such that they did not move or separate relative to each other. This facilitates seamless load transmission from one entity to another. Bonded contact was also used to model the connection between the fibula and fibular crest of the tibiotarsus, even though their respective interfaces were not in actual direct contact.

### Material properties

All entities were modelled as solid, isotropic, linearly elastic materials. Three different materials were defined for the entities being modelled (Table 3): bone, cartilage (for the hip

[1] In an equilateral triangle of edge length $L$, its area is given by $\frac{\sqrt{3}}{4} \cdot L^2$; setting this equal to average triangle area $\frac{A}{n}$ gives Eq. (3).

**Table 3 Material properties used in the finite element analysis component of the simulations.**

| Material | Density (kg/m$^3$) | Young's modulus (MPa) | Poisson's ratio |
|---|---|---|---|
| Bone | 2,060 | 17,000 | 0.3 |
| Cartilage | 1,100 | 50 | 0.45 |
| Knee soft tissues* | 1,100 | 100 | 0.3 |

**Notes:**

All entities were modelled as solid, isotropic, linearly elastic materials. Values derived from *Reed & Brown (2001)*, *Currey (2002)*, *Erickson, Catanese & Keaveny (2002)*, *Stops, Wilcox & Jin (2012)* and *Kazemi, Dabiri & Li (2013)*, and references cited therein.

* The knee soft tissues material properties reflected a composite of those of both cartilage and menisci.

and ankle soft tissue entities) and a composite of the material properties of cartilage and menisci (for the knee soft tissue entity). Extinct, non-avian theropods are inferred to have had menisci in their knee joints, based on their widespread occurrence in extant tetrapods, including birds and crocodilians (*Chadwick et al., 2014*; *Haines, 1942*; *Wink et al., 1989*; *Zinoviev, 2010*), but the actual morphology of these menisci remains speculative. This is one of the reasons for modelling all soft tissues in the knee joint as a single, homogenous entity, in addition to being a computational simplification.

The material properties assigned for bone were conservatively estimated from the most common values reported for cortical bone in the literature (e.g. *Currey, 2002*, and references cited therein; *Erickson, Catanese & Keaveny, 2002*; *Reed & Brown, 2001*). The material properties for cartilage and menisci were also conservative estimates, derived from the literature (*Currey, 2002*, and references cited therein; *Kazemi, Dabiri & Li, 2013*; *Stops, Wilcox & Jin, 2012*).

In previous finite element studies, cartilage and menisci have been represented with a variety of material behaviours, including isotropic and transversely isotropic linear elasticity, hyperelasticity, viscoelasticity and poroelasticity (*Kazemi, Dabiri & Li, 2013*; *Stops, Wilcox & Jin, 2012*). The use of isotropic, linearly elastic material behaviour in the present study is justified on the following grounds. Firstly, as the analyses of the present study were quasi-static, the time (strain rate) dependency of non-linear material properties can be ignored with minimal error (*Carey et al., 2014*). Secondly, the precise kind of material behaviour, or material properties, is virtually unknown for any archosaur (extinct or extant). Thirdly, assuming an isotropic, linearly elastic material behaviour kept the model simple and minimally speculative, and also reduced the computational cost of solving the finite element models.

A solid, isotropic, linearly elastic continuum representation was also necessitated for the bone entities in the simulations. Not only is this because material properties (and any anisotropy thereof) are unable to be determined for extinct theropods, but moreover anything other than this representation could compromise the objectives of the current study. Specifically, the introduction of any sort of structural or material heterogeneity, discontinuity or directionality will influence the resulting principal stress trajectories. Since a key objective of this study was to examine the spatial variation of the calculated principal stress trajectories in relation to cancellous bone architecture, directionality needed to be a model output only, not a model input.

### Loads and restraints

For each simulation, four sets of loads were applied: muscle and ligament forces, joint forces, joint moments and segment weight. As noted above, muscle and ligament forces were applied to the same nodes as were involved in the musculoskeletal simulations. Additionally, a given muscle or ligament force was evenly spread out over a number of nodes (generally around 20), centred about the focal node, in order to reduce the incidence of artifacts in the model results.

Joint forces were applied to a focal bone via its neighbouring bones. Here, one neighbouring bone was restrained in translation in all three axes, whilst the other was used to apply a joint force; the joint force at the restrained end of the bone was provided by the reaction at the restraints, transmitted back through the bone of interest. In both the femur and tibiotarsus + fibula simulations, the knee joint force was applied directly via the appropriate neighbouring bone (proximal tibiotarsus + fibula and distal femur, respectively), with the other neighbouring bone being restrained (acetabulum and proximal tarsometatarsus, respectively). In ANSYS, this approach was implemented by using a 'remote force' (Fig. 3D). This is where a force is applied to a specific entity, but via a remote point in space that is topologically attached ('scoped') to the entity; when a force is applied to the remote point, the target entity gets pulled or pushed along with it, along the line of action of the applied force. In ANSYS, this is accomplished by a set of constraint equations that relate the degrees of freedom of an entity's nodes to the remote point; one constraint equation exists for each node in the entity experiencing the remote force. The location of the remote point in both the femur and tibiotarsus + fibula simulations was specified as the location of the knee joint centre in the musculoskeletal model. This meant that the joint force was applied properly, without introducing any moments into the system, because the net force vector passed through the correct location in space, again ensuring complete correspondence between the finite element and musculoskeletal simulations.

The knee joint moment was applied directly to the appropriate bone or bones, by applying it to the surface or surfaces in contact with the knee soft tissues; for example, by applying it to the distal femur in the femur simulation. This direct application was chosen, as opposed to the moment being applied via a neighbouring bone, because the greater compliance of the knee soft tissues would not allow full transmission of the moment to the bone or bones of interest. No hip joint moment was involved, since the hip joint was modelled as a ball-and-socket joint, and thus unable to resist moments. Whilst an ankle joint moment was calculated in the musculoskeletal simulations, it was not able to applied in the tibiotarsus + fibula finite element simulations. This is because of the close proximity of the ankle end of the tibiotarsus to the restraint, and thus the restraint would greatly alter the transmission of any applied moment; this modelling deficiency will be returned to in the Discussion (section 'Successes and pitfalls').

The weight of the appropriate segment (e.g. thigh segment weight in the femur simulation) was applied via a remote point that was scoped to the entire bone of interest. The location of the remote point was set as the COM of the limb segment.

### Model solution

All finite element simulations were solved as linear static systems in the Static Structural module of ANSYS. Additionally, all simulations used inertia relief, which is a technique that is used to counter unbalanced forces, so as to produce no net acceleration of the model (*Liao, 2011*). This is achieved through the application of an inertial force and moment to the model's centre. Although the musculoskeletal simulations described above were analysed under the assumption of static equilibrium, this does not exactly occur in finite element simulations due to non-rigid behaviour of the various entities. In particular, the soft tissue structures are highly compliant relative to the bone structures, and deformation of these soft tissue structures during simulation will lead to an imbalance of the applied forces. This has the potential to produce a positive-feedback loop where force imbalance leads to model acceleration, which leads to further deformation, which in turn leads to greater force imbalance, and so on. Ultimately, very large and unrealistic deformations occur, and calculated model results are unreliable. Thus, inertia relief was used to counter the initially very small imbalance in forces that results upon deformation of the model; for instance, in the femur simulation of the solution posture, the applied inertial force was

$$(F_x, F_y, F_z) = (7.1725 \times 10^{-8}, 6.7934 \times 10^{-8}, -1.1303 \times 10^{-6}) \text{N},$$

and the applied inertial moment was

$$(M_x, M_y, M_z) = (1.7224 \times 10^{-6}, 3.3496 \times 10^{-6}, -5.3604 \times 10^{-7}) \text{Nm}.$$

The very small magnitude of these adjustments justifies the use of this technique in the current study.

### Results analysis

Upon performing the finite element simulations for a given test posture, the calculated stress tensor at each node in each bone entity was exported from ANSYS. A custom script in MATLAB 8.0 (MathWorks, Natick, MA, USA) was then used to perform an eigenanalysis of the stress tensor data, producing the vector orientations of the principal stresses. Their 3D trajectories were then visualized using this MATLAB script, as well as Rhinoceros. These trajectories, particularly of the maximum principal stress ($\sigma_1$, usually signifying tension) and minimum principal stress ($\sigma_3$, signifying compression) were visually (qualitatively) compared to the architectural patterns of cancellous bone fabric reported for birds in Part I (*Bishop et al., 2018b*). As a further aid to assessing the degree of correspondence between principal stresses and cancellous bone fabric, the direction of $\sigma_3$ in the femoral head and medial femoral condyle was quantitatively compared to the mean directions of the primary fabric direction ($\mathbf{u}_1$) for those parts of the femur in birds, also as reported in Part I. As $\sigma_3$ is compressive, it stands to reason that this will show the greatest correspondence with the architecture in the femoral head and medial femoral condyle, both of which would be expected to be exposed predominantly to compressive joint loading. The direction of $\sigma_3$ in the femoral head was taken to be the mean direction of vectors in the region of a sphere of radius one-half of that fit to the entire femoral

head (performed in 3-matic), and positioned just under the surface of the bone, underneath where the hip force was received in the finite element simulations. The direction of $\sigma_3$ in the medial femoral condyle was taken to be the mean direction of vectors in the region of a sphere of radius one-third of that fit to the condyle (performed in 3-matic), and positioned in the anatomical centre of the condyle. Higher priority was given towards improving correspondence in the femoral head over the medial condyle, since hip angles are presumed to be more important for determining overall posture in bipeds; as the most proximal joint (and with three degrees of freedom), the hip will have the greatest influence on overall limb positioning (cf. *Hutchinson & Gatesy, 2000*), as well as the disposition of more distal joints in the limb. Additionally, strict comparison between the mean directions of $\sigma_3$ and $\mathbf{u}_1$ in the medial femoral condyle ignores the 'fanning' of $\mathbf{u}_1$ that occurs in this region of the bone (see Part I), and hence is less legitimate.

Comparisons were made from the chicken finite element stress results to the architectural patterns observed in ground-dwelling birds as a whole for two main reasons:

1. It has been shown that birds as a whole appear to demonstrate a largely consistent pattern of cancellous bone architecture in the femur, tibiotarsus and fibula (Part I). That is, the architectural patterns thus far observed can be described by a single 'archetype', about which there was specimen-specific (perhaps species-specific, pending greater sampling) variation.
2. Cancellous bone architecture could not be extensively quantified in smaller birds such as chickens, owing to continuum level restrictions (relatively few trabeculae; see Part I). Nevertheless, where cancellous bone fabric was quantifiable in chickens, the results were close to the mean 'archetypal' value for birds overall (Part I).

As this series of studies is exploratory with a small sample size for each examined avian species (Part I), it is prudent (and conservative) to make comparisons to ground-dwelling birds as a whole, until such a time that significant interspecific differences can be demonstrated, in terms of both locomotor behaviour and cancellous bone architecture.

## Caveats

Two points are worth noting about the overarching philosophy of the approach of the current study. Firstly, this study sought to determine a *single posture*, whose principal stress trajectories showed the greatest degree of correspondence to observed cancellous bone architecture in the femur, tibiotarsus and fibula. Cancellous bone, however, experiences many different loading regimes throughout the course of normal activity, each of which engenders a remodelling stimulus, and to which cancellous bone responds and adapts its architecture (*Kivell, 2016*). This has been demonstrated in many previous computational theoretical studies, whereby no one loading regime will lead to replication of all of the observed architectural features in a bone; only when multiple loading regimes are considered can all of a bone's cancellous architecture be explained (*Beaupré, Orr & Carter, 1990*; *Bona, Martin & Fischer, 2006*; *Boyle & Kim, 2011*; *Carter & Beaupré, 2001*; *Carter, Orr & Fyhrie, 1989*; *Coelho et al., 2009*; *Jacobs et al., 1997*;

*Jang & Kim, 2008*, *2010a*, *2010b*; *Phillips, Villette & Modenese, 2015*; *Sverdlova, 2011*; *Tsubota, Adachi & Tomita, 2002*; *Tsubota et al., 2009*; *Turner, Anne & Pidaparti, 1997*). Therefore, in seeking a single posture that best reflects the observed cancellous bone architecture, the current study in fact searched for a 'characteristic posture', which is a time- and load-averaged posture across all loading regimes. This characteristic posture may or may not be an actual posture used at a particular instance in a particular behaviour. As argued above, however, the posture at around the mid-stance of a stride will probably be important, and the characteristic posture so derived may therefore bear considerable resemblance to it.

Secondly, many assumptions and modelling simplifications were necessarily made in this study. Many of these were necessitated by the lack of empirical data for extinct, non-avian theropods, such as soft tissue anatomy or material properties, which in all likelihood will never be obtained. Other simplifications pertained to making the system more tractable for analysis and interpretation, such as representing the knee and ankle joints with a single degree of freedom each, when it is known that these joints are capable of more complex motions during locomotion in birds (*Kambic, Roberts & Gatesy, 2014*; *Rubenson et al., 2007*). All of the assumptions and simplifications involved in the present study could in principle be investigated via sensitivity analysis, but no such analysis was performed here, save for the assumption of constant maximum force across all musculotendon actuators (see next section below). All other assumptions were kept at their 'best guess' manifestation throughout the study. By keeping every aspect of every stage of the modelling process constant, and only varying posture, this allowed for the direct comparison of simulation results to postural differences: differences in model results were entirely due to differences in limb posture. When these assumptions are also held constant in a comparative context across species (*Bright, 2014*), this also allows for a more direct assessment of the effects of posture on limb bone loading and muscular recruitment (Part III).

## Sensitivity to muscle forces

In the musculoskeletal simulations, all musculotendon actuators were assigned the same maximum force (2 BW) for the sake of simplicity and also to facilitate consistency across extinct and extant theropod species. In reality, the varying sizes and architectures of the different muscles mean that they can have greatly different maximal force capabilities, which may have an important effect on the end results. To examine how sensitive the results were to using more realistic muscle force capabilities, the solution posture identified above was re-analysed with muscle-specific maximum force capacities stipulated. The original chicken carcass used to build the model was not able to be dissected for measurement of muscle architecture, and so the data collected by *Paxton et al. (2010)* for an adult (2.08 kg) junglefowl were used instead, scaled to the chicken model in proportion to mass. Maximum muscle force for each of the active muscles was then calculated following standard formulae, assuming a constant isometric stress of $3 \times 10^5$ N/m$^2$ (*Medler, 2002*); values are reported in Table 4. The maximum force for the four collateral ligaments modelled was left unaltered from the original values.

**Table 4  Maximum force capability of the active muscle actuators used in the muscle force sensitivity test.**

| Muscle | $F_{max}$ (N) |
|---|---|
| ILPO | 30.111 |
| AMB | 1.112 |
| FMTE | 19.636 |
| FMTM | 21.007 |
| FMTI | 92.110 |
| ILFB | 24.777 |
| FCLP | 20.760 |
| FCLA | 18.656 |
| FCM | 8.713 |
| ITM | 3.003 |
| ITC | 77.382 |
| ISF | 25.635 |
| CFC | 1.625 |
| CFP | 6.300 |
| PIFL | 7.981 |
| PIFM | 17.940 |
| GL | 59.539 |
| GI | 10.863 |
| GM | 71.969 |
| FDL | 31.022 |
| ODF | 58.122 |
| FHL | 22.672 |
| FL | 51.621 |
| FB | 8.737 |

**Note:**
Note how the force capability can vary widely, from less than 0.1 BW (AMB) to over 6 BW (FMTI).

The results for both the musculoskeletal and finite element simulations of this sensitivity test are reported below alongside those for the main simulations.

## RESULTS

A total of eight different postures were tested before no closer correspondence between principal stress trajectories and cancellous bone architectural patterns was achieved. This was determined from both the qualitative visual inspection of stress trajectories and architectural patterns in all three bones, as well as quantitative minimization of angular deviation between stresses and fabric directions in the femoral head and medial femoral condyle. Going from the worst to best postures tested, the angular deviation between the minimum principal stress ($\sigma_3$) and the primary fabric orientation ($u_1$) in the femoral head decreased from 23.3° to 7.9°, a 66% reduction; likewise, the angular deviation between $\sigma_3$ and $u_1$ in the medial femoral condyle decreased from 29.2° to 17.3°, a 41% reduction. The final solution posture is illustrated in the centre of Fig. 4. The 'degree of crouch' (*Bishop et al., 2018*) of this posture is 0.160; the degree of crouch in a standing

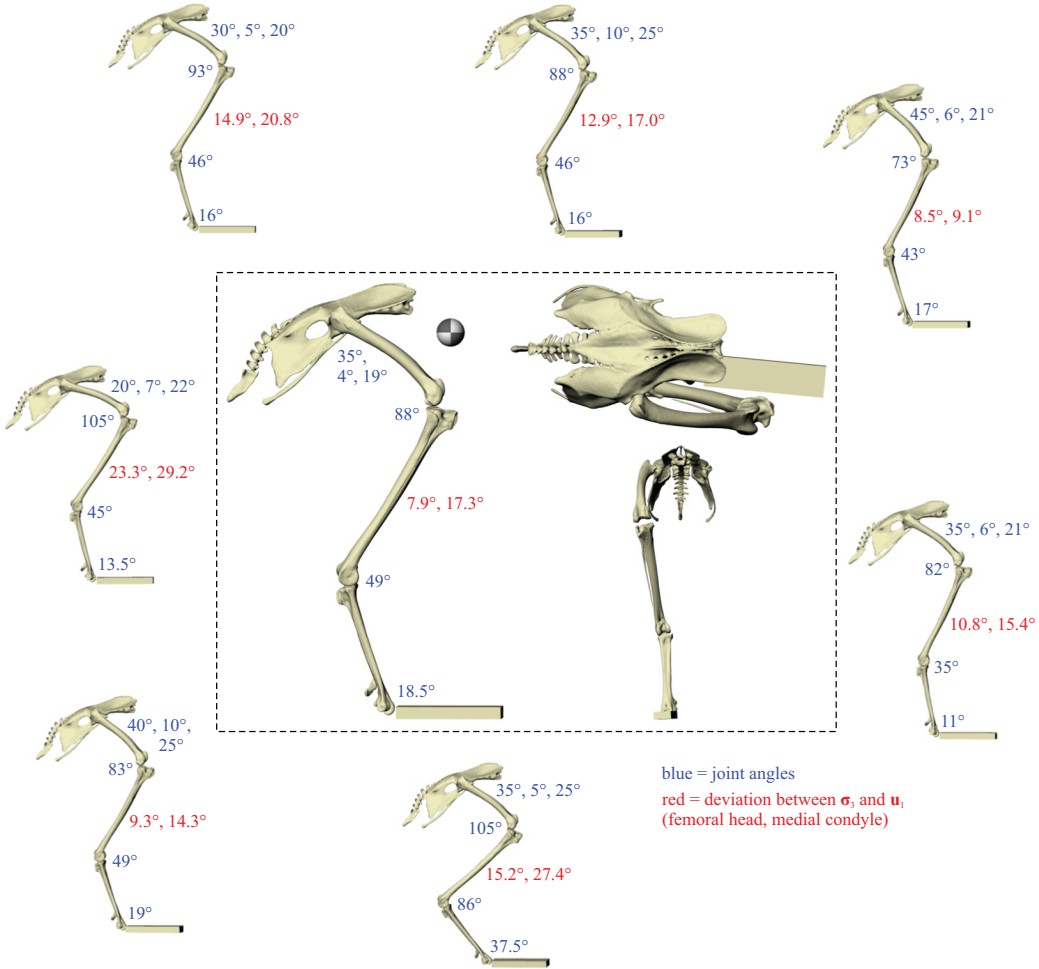

**Figure 4 The postures tested for in the chicken.** Around the periphery are the different postures tested, shown in lateral view, with the final solution posture in the centre box, shown in lateral, dorsal and anterior views; the whole-body COM location is also shown for the solution posture in lateral view. Joint angles for each posture are given in blue font; hip joint angles are given in the order of flexion–extension, abduction-adduction and long-axis rotation. Hip extension angle is expressed relative to the horizontal, whereas knee and ankle angles are expressed relative to the femur and tibiotarsus (respectively). For the other hip angles, positive values indicate abduction and external rotation, whereas negative values indicate adduction and internal rotation. The metatarsophalangeal joint angle is expressed relative to the neutral posture. The angular deviation between $\sigma_3$ and $u_1$ for each posture is also given in red font (reported as femoral head, then medial femoral condyle). The solution posture resulted in the greatest degree of overall correspondence between principal stress trajectories and observed cancellous bone architectural patterns in birds, as assessed by qualitative comparisons across the femur, tibiotarsus and fibula, as well as quantitative results for the femoral head and medial femoral condyle.

posture, as empirically predicted from the total leg length of the chicken individual modelled (275 mm), would be 0.166 (*Bishop et al., 2018*). It is worth remembering that despite this close similarity, the solution posture should not be equated literally with any single real posture used (be it of standing, slow walking, fast running, etc.), for it is a characteristic weighted average of all postures used.

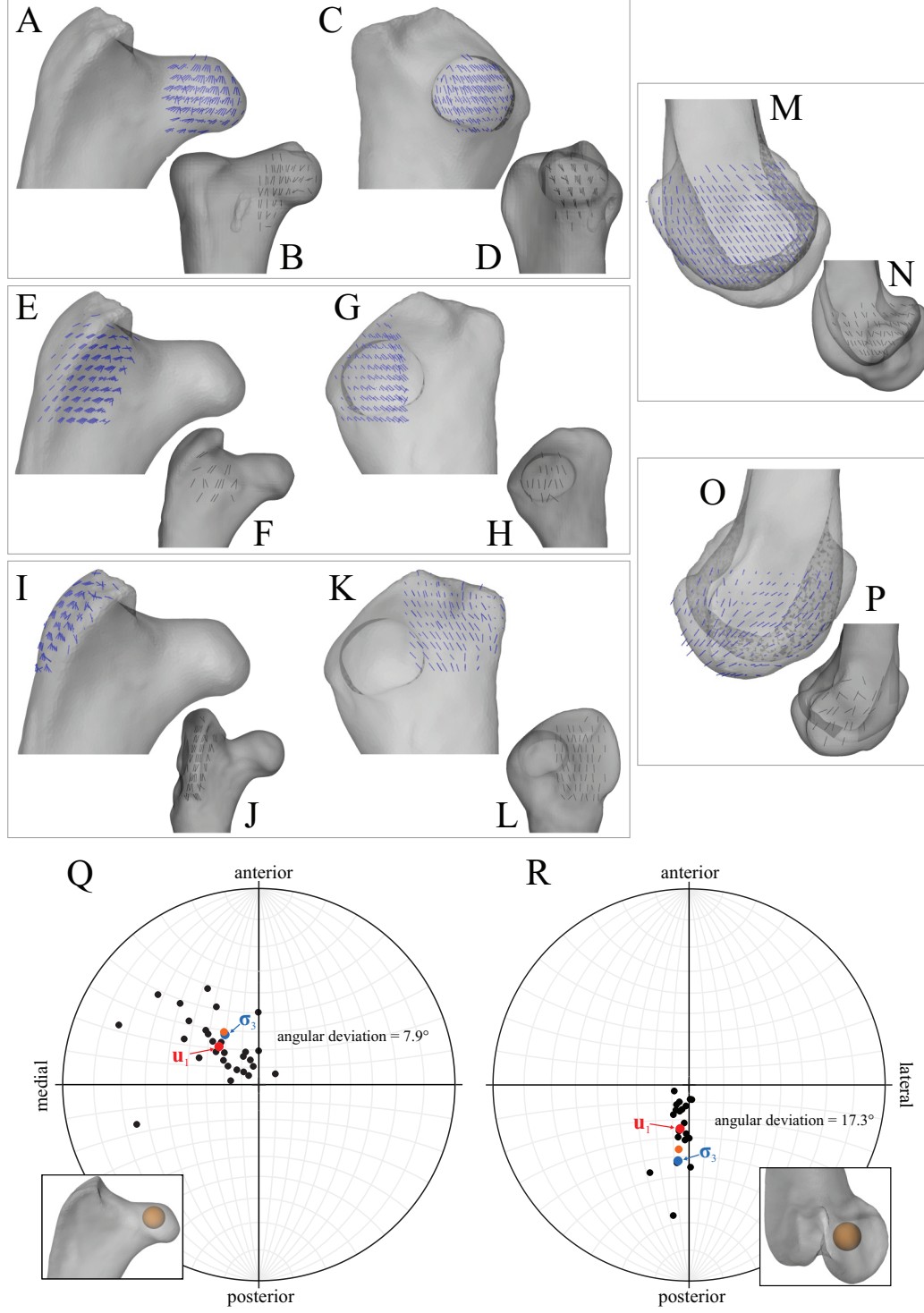

**Figure 5 Principal stress trajectories for the femur in the solution posture compared to cancellous bone fabric.** (A, C, E, G, I, K, M, O) Stress vector fields ($\sigma_3$ in all cases) compared to exemplar fabric vector fields for birds (B, D, F, H, J, L, N, P, $u_1$ in all cases; cf. Figs. 16 and 24 of Part I), plotted on translucent renderings of the bone; not to scale. For easier visual comparison, the stress trajectories were 'downsampled' in a custom MATLAB script, by interpolating the raw stress results at each finite element node to a regular grid. (A–D) In the femoral head, in anterior (A, B) and medial (C, D) views.

(E–H) Under the facies antitrochanterica, in anterior (E, F) and lateral (G, H) views. (I–L) In the trochanteric crest, in anterior (I, J) and lateral (K, L) views. (M, N) Medial femoral condyle, parallel to the sagittal plane and in medial view. (O, P) Lateral femoral condyle, parallel to the sagittal plane and in lateral view. (Q) Comparison of the mean direction of $\sigma_3$ (blue) in the femoral head and the mean direction of $u_1$ (red) for birds, plotted on an equal-angle stereoplot, with northern hemisphere projection (using StereoNet 9.5; *Allmendinger, Cardozo & Fisher, 2013*; *Cardozo & Allmendinger, 2013*). (R) Comparison of the mean direction of $\sigma_3$ in the medial femoral condyle and the mean direction of $u_1$ for birds, plotted on an equal-angle stereoplot, with southern hemisphere projection. Insets in (Q) and (R) show locations of regions for which the mean direction of $\sigma_3$ was calculated. The orange dots in (Q) and (R) indicate the mean direction of $\sigma_3$ for the muscle force sensitivity test; note how close these are to the original results for the solution posture.               

### Principal stress trajectories

In the solution posture, the principal stress trajectories in the femur, in particular those of $\sigma_3$ (compressive), showed a high degree of correspondence with the observed cancellous bone architectural directions, in the femoral head, under the facies antitrochanterica, in the trochanteric crest and in both femoral condyles (Figs. 5A–5P). The mean direction of $\sigma_3$ in the femoral head showed strong correspondence to the mean direction of $u_1$ measured for birds (Fig. 5Q); compared to just the mean direction for the chickens studied in Part I ($n = 2$), the mean direction of $\sigma_3$ had the same general azimuth, but was less proximally inclined (angular deviation of 18°). Fair correspondence between $\sigma_3$ and $u_1$ also occurred in the medial femoral condyle, although the direction of $\sigma_3$ was notably more posteriorly inclined than the mean direction of $u_1$ across all birds (Fig. 5R); a more posteriorly inclined orientation of $\sigma_3$ occurred in all postures tested.

Much correspondence between principal stress trajectories and cancellous bone architecture was also observed in the tibiotarsus, particularly in the proximal end (Fig. 6). In the anterior cnemial crest, the trajectory of the maximum principal stress ($\sigma_1$, tensile) largely paralleled the margins of the crest, as observed for cancellous bone fabric. In much of the lateral cnemial crest, the observed cancellous bone fabric reported for birds was reflected by the trajectory of $\sigma_3$. Under the articular facies, the trajectory of $\sigma_3$ corresponded closely with the observed architectural patterns there, showing a posterior inclination largely parallel to the sagittal plane. Additionally, in the sagittal plane through the middle of the proximal end, $\sigma_1$ and $\sigma_3$ formed a double-arcuate pattern, closely resembling a similar pattern in $u_1$ observed in some of the large bird individuals studied in Part I (Figs. 6S and 6T). In contrast to the proximal tibiotarsus, only minimal correspondence between principal stress trajectories and cancellous bone architecture could be attained in the distal tibiotarsus, in any posture tested. In the solution posture, there was some correspondence between $\sigma_3$ and observed architecture in the immediate vicinity of the articular condyles, where $\sigma_3$ was largely parallel to the sagittal plane (Figs. 6U and 6V), but this was not observed throughout the entire distal end of the bone, unlike the architecture.

The principal stress trajectories in the fibular head showed strong correspondence to the gentle inclination observed in the cancellous bone architecture (Fig. 7). Medially, $\sigma_1$ showed this pattern, whereas laterally, it was $\sigma_3$ that showed this pattern.

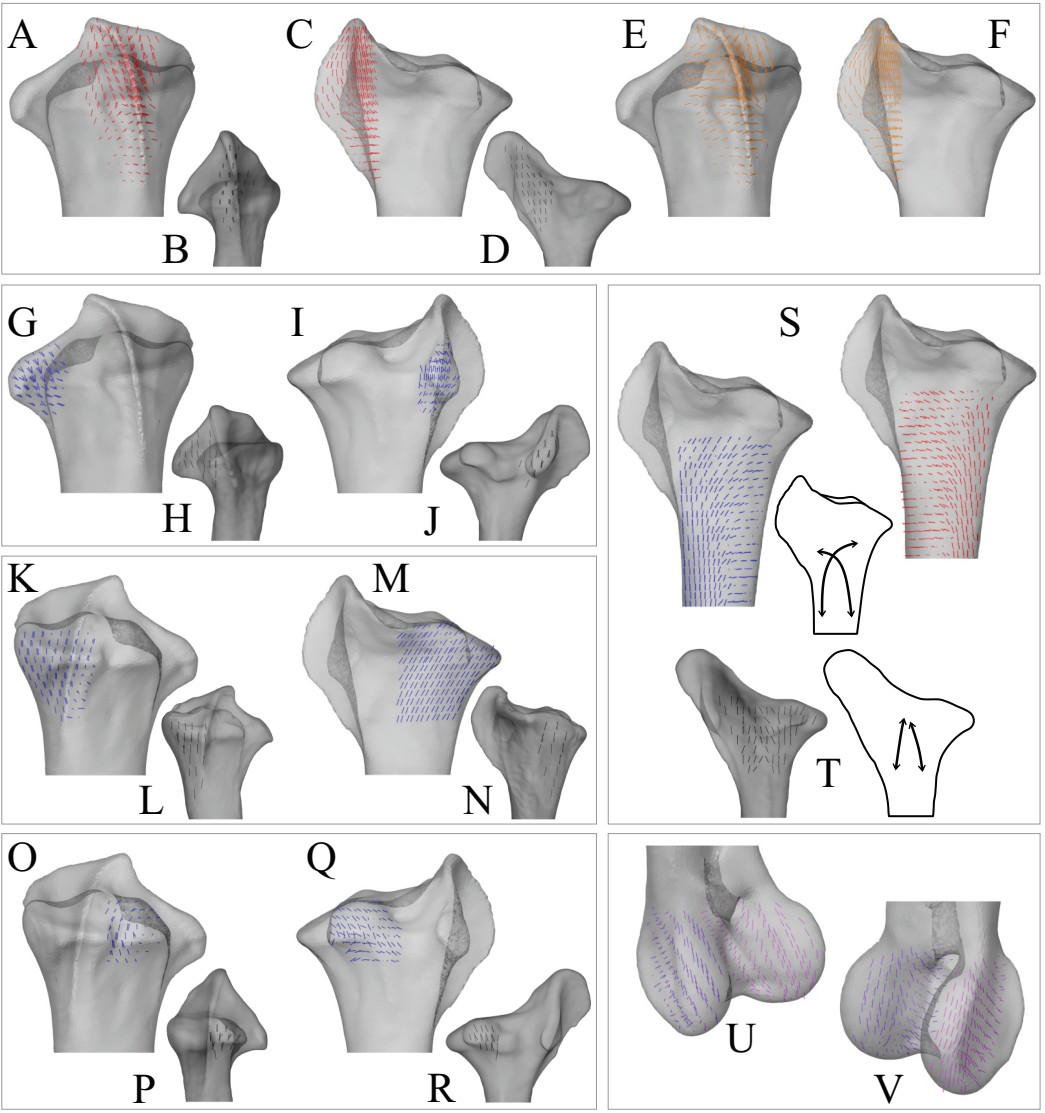

**Figure 6 Principal stress trajectories for the tibiotarsus in the solution posture compared to cancellous bone fabric.** (A, C, G, I, K, M, O, Q, S) Stress vector fields ($\sigma_1$ in red, $\sigma_3$ in blue) compared to exemplar fabric vector fields for birds (B, D, H, J, L, N, P, R, T, $\mathbf{u}_1$ in all cases; cf. Figs. 31 and 36 of Part I), plotted on translucent renderings of the bone; not to scale. (A–D) Anterior cnemial crest, in anterior (A, B) and medial (C, D) views. (E, F) Vector field of $\sigma_1$ in the anterior cnemial crest in the muscle force sensitivity test, shown in the same views as A and C, respectively. (G–J) Lateral cnemial crest, in anterior (G, H) and lateral (I, J) views. (K–N) Under the medial articular facies, parallel to the coronal plane (K, L, posterior view) and sagittal plane (M, N, medial view). (O–R) Under the lateral articular facies, parallel to the coronal plane (O, P, posterior view) and sagittal plane (Q, R, lateral view). (S, T) A 3D slice through the middle of the proximal metaphysis, parallel to the sagittal plane; schematic insets show the double-arcuate pattern present in both the stress trajectories and fabric vectors. (U, V) Vector field of $\sigma_3$ in the articular condyles (purple = lateral condyle, pink = medial condyle) of the distal tibiotarsus, shown for 3D slices through the middle of the condyles, in oblique anterolateral (U) and anteromedial (V) views.                               

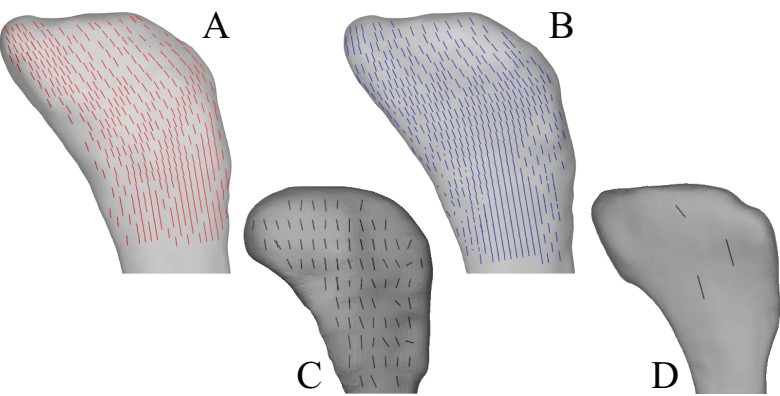

**Figure 7 Principal stress trajectories for the fibula in the solution posture compared to cancellous bone fabric.** (A) Vector field of $\sigma_1$ in the medial side of the fibular head plotted on a translucent rendering of the bone, in medial view (reversed). (B) Vector field of $\sigma_3$ in lateral side of the fibular head, in lateral view. (C, D) Exemplar fabric vector fields ($\mathbf{u}_1$) for birds, in lateral view (cf. Figs. 40G–40K of Part I); not to scale.           

## Mid-shaft stresses

In the solution posture, the most axis-parallel orientation of both $\sigma_1$ and $\sigma_3$ at the femoral mid-shaft was at a high angle to the long-axis of the bone, by at least 30°, indicating considerable torsion (Fig. 8A). Moreover, the sense of torsion as indicated by the stress trajectories was positive; when the right femur is viewed proximally, the proximal end was rotated counterclockwise relative to the distal end (Fig. 8B). The neutral surface of bending was oriented 36° from the mediolateral axis, indicating that bending of the femur was predominantly in an anteroposterior direction.

In the tibial mid-shaft, the most axis-parallel orientation of $\sigma_1$ and $\sigma_3$ was almost parallel to the long-axis of the bone, indicating only a minimal torsion (Fig. 8C). The sense of torsion (what very little there is) as indicated by the stress trajectories was also positive. The neutral surface of bending was oriented 19° from the mediolateral axis, indicating that bending of the tibiotarsus was also in a predominantly anteroposterior direction.

## Muscle and ligament activations

In the solution posture, the activations of the four collateral ligament actuators were very low (0.012 or less), indicating that the vast majority of limb stabilization, excluding the metatarsophalangeal joint, was conferred by muscle actuators. However, as the knee and ankle were represented as hinge joints in this study, joint stabilization would also have been achieved in part through resistance offered by these single degree-of-freedom joints to off-axis moments and forces. (Indeed, this resistance could well be responsible for the minimal recruitment of ligaments in the first place.) This resistance was nevertheless transmitted to the bones as joint moments and forces (calculated in the musculoskeletal simulations). Therefore, as far as the bones are concerned, all experienced loads are accounted for and incorporated into the finite element simulations. Nevertheless, as off-axis loads were able to be partly resisted by the hinge joints, the calculated forces in the collateral ligaments may be appreciably less than what they would be *in vivo*.

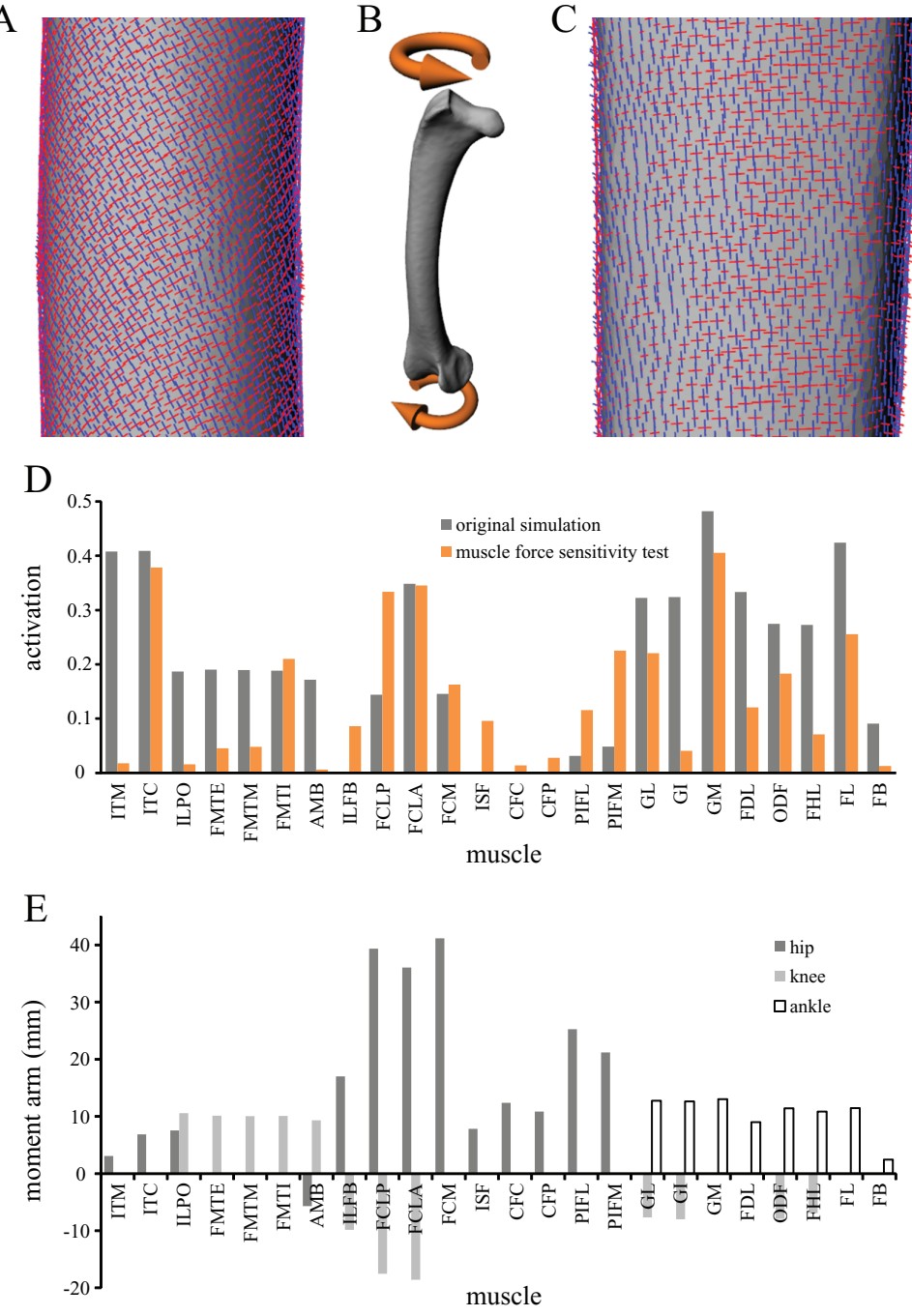

**Figure 8 Additional aspects of the solution posture.** (A) The trajectories of the principal stresses $\sigma_1$ (red) and $\sigma_3$ (blue) at the femoral mid-shaft, in anterior view. (B) Oblique principal stresses in the femoral mid-shaft indicate of strong torsional loading (orange arrows), with a positive sense. (C) The trajectories of $\sigma_1$ and $\sigma_3$ at the tibial mid-shaft, in anterior view; note the almost complete lack of obliquity with respect to the long-axis of the bone. (D) Activations for each muscle actuator in the musculoskeletal simulation. In the original simulation all musculotendon actuators were assigned a maximum force of two body weights, whilst in the sensitivity test the assigned maximum force was specific to the size and architecture of each muscle. (E) Flexion–extension muscle moment arms for the hip, knee and ankle joints; positive values indicate extension, negative values indicate flexion. For keys to abbreviations in (D and E), see Table 1.     

The activations of all muscle actuators are presented in Fig. 8D. Most muscles were recruited with activations above 0.1 (i.e. clearly active); no muscle was recruited beyond half of its maximal capacity. The iliotrochanterici medius (ITM) et caudalis (ITC), gastrocnemius medialis (GM) and FL were the most recruited muscles, each with an activation above 0.4. The iliofibularis (ILFB), ischiofemoralis (ISF) and caudofemoralis partes caudalis (CFC) et pelvica (CFP) were the least recruited muscles, all with negligible activations.

**Muscle force sensitivity test**

The activations of muscle actuators in the sensitivity test were sometimes markedly different from those of the original simulation (Fig. 8D). However, there was also agreement between the two simulations, where several key hindlimb muscles were predicted to be important in both cases (e.g. ITC, FMTI, GM, GL and FL). Despite the differences in muscle maximum force capabilities between the original and sensitivity test simulations (Table 4), as well as the differences in calculated muscle activations, the stress results differed little from those of the original simulation. The qualitative patterns of stress trajectories were highly similar to those of the original simulation in most regions of the femur, tibiotarsus and fibula. The only region where there was a marked difference was in part of the anterior cnemial crest, in terms of $\sigma_1$ (Figs. 6E and 6F, cf. Figs. 6A and 6C). In terms of quantitative results, there was again little deviation in the mean direction of $\sigma_3$ from that calculated for the original simulation (Figs. 5Q and 5R, orange dots). The difference in mean $\sigma_3$ directions between the original and sensitivity test simulations was very small in both the femoral head (1.3°) and medial femoral condyle (5.6°). These results suggest that the approach of assigning a single constant value of 2 BW for muscle maximum force capacity does not introduce a significant degree of error, at least as far as the objectives of the present study are concerned.

# DISCUSSION

The aim of this study was to verify the 'reverse' application of the trajectorial theory, to go from observed cancellous bone architectural patterns to bone loading regimes and limb postures, as applied to theropod locomotor biomechanics. This was achieved through the development of a novel approach that integrated musculoskeletal and finite element simulations of a modern theropod, the chicken. By focusing on a modern theropod here, the validity of the reverse approach was able to examined for each major bone in the hindlimb.

**Successes and pitfalls**

Despite the many modelling simplifications made in the current study, and that only a single static posture was modelled for any one test, much of the observed cancellous bone architectural patterns in the avian hindlimb was able to be replicated in the principal stress trajectories. This was particularly true of the femur. The 'solution posture' that produced the greatest correspondence between principal stresses and cancellous bone architecture is qualitatively comparable to the hindlimb posture of medium-sized birds such as chickens and guineafowl (*Carrano & Biewener, 1999*; *Gatesy, 1999a*; *Grossi et al.,*

2014), and less comparable to the postures of larger birds with more extended joints (*Abourachid & Renous, 2000*; *Rubenson et al., 2007*) or smaller birds with more flexed joints (*Reilly, 2000*; *Stoessel & Fischer, 2012*). Furthermore, the degree of crouch in the chicken model was almost identical to what would be empirically predicted based on limb bone lengths, for a quiet standing posture (*Bishop et al., 2018*). It is important to reiterate that the solution posture obtained here is a weighted average (by both time and load) of all postures and loading regimes experienced on a daily basis, and yet the relative weightings of each posture and loading regime are not actually known. Hence, it would be worthwhile in future studies to more exactly discern what aspects of locomotor biomechanics the solution posture most strongly reflects.

Other aspects of the solution posture also showed correspondence with empirical data for avian terrestrial locomotion. The femur was predicted to be loaded in considerable torsion, with a positive sense, as well as bending that was predominantly anteroposteriorly directed. This is consistent with the loading regimes recorded by *in vivo* strain gauge studies of chickens and emus (*Carrano, 1998*; *Carrano & Biewener, 1999*; *Main & Biewener, 2007*). Additionally, the two most strongly recruited muscles in the musculoskeletal simulations, the GM and FL, are also the two largest muscles in the distal hindlimb of birds (*Lamas, Main & Hutchinson, 2014*; *Paxton et al., 2010*; *Smith et al., 2006*), and would therefore be expected to be capable of producing large amounts of force, all other factors being equal.

There were also a few aspects in which the solution posture did not accord well with empirical observations. Most pertinently, the principal stress trajectories in the distal tibiotarsus did not show much correspondence with the cancellous bone architecture observed in this region of the bone of birds (Part I). This may suggest that the manner in which that part of the bone was modelled in the current study was inadequate, that is, too non-physiological. For instance, the ankle joint moment calculated in the musculoskeletal simulations was not able to be applied in the finite element simulations in their current formulation. The discrepancy may also be due to the cancellous bone architecture of the distal tibiotarsus reflecting many different loading regimes, any single one of which cannot capture the architecture. Some of those loading regimes may be very different to that occurring around the mid-stance of locomotion, such as those associated with the swing phase of locomotion, or even standing and sitting. Alternatively, the poor correspondence may indicate that the trajectorial theory may not actually hold true here for some reason, potentially related to the ontogenetic fusion of the proximal tarsals and distal tibia of birds (see *Lovejoy, 2004*; *Lovejoy et al., 2002*). A second aspect in which the solution posture did not concur with empirical observations concerned the stresses at the tibial mid-shaft. Here, the bone was predicted to be loaded in only minimal torsion, the sense of which was positive; this does not accord with *in vivo* strain gauge studies, which have shown that the avian tibiotarsus experiences a large amount of torsional loading during locomotion, which furthermore is of a negative sense (*Biewener, Swartz & Bertram, 1986*; *Main & Biewener, 2007*; *Verner et al., 2016*). It is possible that if additional degrees of freedom were assigned to the model (e.g. long-axis rotation in the knee or ankle; *Kambic, Roberts & Gatesy, 2014*), more realistic results may

have been achieved here (see also *Bell, Snively & Shychoski, 2009*, in relation to ornithischian dinosaur jaws).

A final incongruence between the solution posture and empirical observations was the negligible recruitment of some muscles in the static optimization routine of the musculoskeletal simulations. There were four such muscles (ILFB, ISF and caudofemoralis partes caudalis et pelvica), yet electromyography data indicates that at least three of these (ILFB, ISF and caudofemoralis pars caudalis) are active during a significant part of the stance phase (*Gatesy, 1990*, *1999b*; *Jacobson & Hollyday, 1982*; *Marsh et al., 2004*). The negligible recruitment of the ISF and caudofemoralis is consonant with the generally small size of these muscles in birds, but this does not hold for the ILFB, which is quite large (*Hudson, Lanzillotti & Edwards, 1959*; *Lamas, Main & Hutchinson, 2014*; *McGowan, 1979*; *Patak & Baldwin, 1998*; *Paxton et al., 2010*; *Smith et al., 2006*). It is probable that all four muscles were minimally recruited in the static optimization routine on account of (i) all muscle actuators were assigned the same maximum capable force output, and (ii) these four particular muscles have smaller moment arms of hip extension compared to other muscles, such as the flexores crures medialis, lateralis pars pelvica et lateralis pars accessoria (Fig. 8E). That is, the static optimization preferentially recruited muscles with larger moment arms, such that lower forces, and therefore activations, were required to provide the necessary stabilizing joint moments.

The last two aspects of discrepancy between the solution posture and empirical data may also in part reflect the fact that the musculoskeletal models were analysed as quasi-static systems. Dynamic aspects of locomotion, such as inertial forces or relative movement between bones, may lead to increased levels of torsion in the tibiotarsus. The same dynamic effects can also influence the net joint moments required to be stabilized by muscle forces; for instance, active retraction of the hip and flexion of the knee may lead to greater activation of the ILFB. Therefore, the activations calculated in the current study are probably most informative for muscles that predominantly confer postural stability, rather than active limb movement (i.e. those that act as brakes or motors; *Rankin, Rubenson & Hutchinson, 2016*).

## Limitations and future work

Numerous assumptions and modelling simplifications were made in the course of this study, and these have already been discussed in detail above. However, this study had other attendant limitations, which are noted here and which provide the basis for future work. Owing to the constraints of time and resources, it was only feasible to model a single avian species to test the validity of the reverse application of the trajectorial theory. In the context of the present study's objectives, this was deemed sufficient; yet, the modelling of additional species in the future (across the size spectrum of extant birds) could help to further clarify the strengths and weaknesses of the approach. A further limitation of this study concerns the use of quantitative comparisons of principal stresses and cancellous bone fabric, which was restricted to two regions of the femur. This was in part due to time constraints, but it also stemmed from similarly restricted quantification of cancellous bone fabric in Part I of this series. It will be informative in future work to

expand the number of regions of the femur subjected to quantitative comparisons, and expand this to include the tibiotarsus and fibula as well. Such an expanded quantification will likely have to negotiate region-specific obstacles to consistent calculation, such as the anteroposterior 'fanning' of fabric vectors in the distal femoral condyles (Part I). Not only will expanded quantification improve the rigour of analyses and comparisons, but it may also lend itself to the implementation of a more formal (and automated) optimization approach to deriving a solution posture for a given species. Furthermore, the use of a more automated approach may allow additional degrees of freedom to be tractably incorporated into the models (e.g. in the knee; see above).

## Applicability of the reverse trajectorial approach to extinct species

Notwithstanding the aforementioned discrepancies and limitations, the concept of applying the trajectorial theory in reverse is, overall, well supported by the results of the present study. The reverse approach therefore has the potential to provide insight into the loading regimes experienced by extinct, non-avian theropods during locomotion, and more broadly the postures used during locomotion.

Previous studies have sought to use the architecture of cancellous bone to derive loading conditions experienced *in vivo*, although this has largely been confined to theoretical studies of modern animals. Some of these studies have focused on utilizing the spatial distribution of the bulk density of cancellous bone, to which remodelling algorithms (*Bona, Martin & Fischer, 2006*; *Fischer, Jacobs & Carter, 1995*) or artificial neural networks (*Campoli, Weinans & Zadpoor, 2012*; *Zadpoor, Campoli & Weinans, 2013*) are applied to retrieve one or more loading regimes. Presently, these studies have only been implemented in two dimensions, and so their efficacy in analysing complex, 3D geometries or loading regimes (such as torsion) remains to be determined. More importantly for the study of extinct species, however, the process of fossilization will greatly hamper any attempt founded upon the bulk density of cancellous bone. Geological chemical alteration (diagenesis) has the potential to greatly alter the physical density of a fossil bone, and moreover this alteration may not be uniform across a bone, such that it may be impossible to reconstruct the original patterns of bulk density in the living bone.

Diagenesis does not, however, normally alter the actual *structure* of bone; indeed, fine, cellular-level structures are frequently preserved in the fossil bones of dinosaurs and other vertebrates (*Chinsamy-Turan, 2005*; *Houssaye, 2014*). It is the structural characteristics of cancellous bone architecture that are utilized in the present study, namely, fabric directions. Regardless of alterations to bulk density, so long as the actual structure of cancellous bone is preserved in a given fossil, and can be imaged appropriately, then the approach of the present study is feasible. (Of course, bones or regions thereof that have suffered taphonomic deformation should be avoided; *Bishop et al., 2017a*). The structural characteristics of cancellous bone have also been used previously in the deduction of *in vivo* loads, in a series of studies by *Christen et al. (2012*, *2013a*, *2013b*, *2015*). These studies developed voxel-based micro-finite element models that modelled each individual trabecula of a bone, and sought to determine the loading regime, or combination of loading regimes, that achieved the most uniform distribution of strain energy density

 

across the model. The great geometric complexity of the models used in these studies necessitated immense computational capability; only small bones or parts of larger ones were able to be modelled. The computational requirements would quickly become prohibitively large for the modelling of whole bones of even a modest size. Moreover, such geometric complexity would be impossible to produce for medium-sized or large bones, for which high-resolution CT imagery is currently unattainable. An additional problem faced by these computational studies is that currently only very basic loads are able to be examined, and these are only applied at the joints; muscle forces were not considered.

In light of the above discussion, the advantages of the reverse trajectorial approach developed in the current study are clear. Firstly, it is based on the actual structure of cancellous bone, which is usually resistant to alteration by diagenetic processes. Moreover, the structural information required (i.e. fabric directionality) can be ascertained for specimens of a wide range of sizes; each individual trabecula need not be imaged at micron-level resolution, although how the image data is acquired and analysed may vary with the size of the specimen, as demonstrated in Part I. The reverse approach is also easily implemented as a fully 3D analysis that is relatively computationally inexpensive to perform; using a computer with 32 Gb of memory and a 2.4 GHz processor, no single simulation of the present study took more than 10 min to solve. However, the main advantage of the reverse approach is that it explicitly links whole-bone cancellous architecture to whole-limb musculoskeletal mechanics. Thus, cancellous bone architectural patterns can be used to directly test hypotheses of limb posture, muscle control and bone loading mechanics, as will be done in Part III.

## CONCLUSION

In this study a new, mechanistic approach to reconstructing locomotor biomechanics in theropods was developed and tested. Its underlying concept of applying the trajectorial theory in reverse was overall well supported by the results of the present study: cancellous bone architecture can be used to derive bone loading regimes, and in turn limb postures. This is achieved through the integration of 3D musculoskeletal and finite element models with observations of cancellous bone fabric direction.

With just a single, quasi-static posture of a chicken hindlimb, modelled with a number of relatively simple assumptions, a large portion of the observed patterns in cancellous bone architecture in birds was able to be replicated by principal stress trajectories. This posture correlated to those actually used during locomotion in birds, in particular the postures used at around mid-stance of normal terrestrial locomotion. Additionally, other biomechanical aspects of the posture, including loading mechanics of the femur and the activations of certain muscles, corresponded well to empirical data recorded for birds. Less agreement between principal stresses and cancellous bone fabric was achievable in the distal tibiotarsus, which requires further investigation, possibly involving more complex modelling approaches (e.g. incorporating more degrees of freedom, or incorporating dynamic effects such as segment accelerations and relative movement between bones).

The reverse approach therefore holds great promise for better understanding whole-bone and whole-limb musculoskeletal biomechanics in the hindlimbs of non-avian

theropods during terrestrial locomotion. The generality of this approach also means that it could also be used to improve understanding of locomotor biomechanics in other extinct tetrapod vertebrate groups as well. As correspondence between principal stresses and cancellous bone architecture was greatest in the femur in the present study, this suggests that the reverse approach will yield the most insight for more proximal limb segments.

## ACKNOWLEDGEMENTS

The staff of the Geosciences Program of the Queensland Museum is thanked for the provision of workspace and access to literature: A. Rozefelds, K. Spring, R. Lawrence, P. Tierney, J. Wilkinson and D. Lewis. Much appreciation is extended to N. Newman (Queensland X-ray, Brisbane) and R. Lawrence for assistance with CT scanning. The thorough and constructive comments on earlier versions of the manuscript, provided by S. Gatesy, T. Ryan, D. Henderson, E. Snively and an anonymous reviewer, are all greatly appreciated, and substantially improved the clarity and content of the research presented here.

### Funding

This study was financially supported by an Australian Government Research Training Program Scholarship (to Peter Bishop), the Paleontological Society (Robert J. Stanton & James R. Dodd Award, to Peter Bishop), an International Society of Biomechanics Matching Dissertation Grant (to Peter Bishop), an Australian Research Council DECRA Fellowship (DE120101503, to Christofer Clemente) and the donation of CT scan time and technical assistance by Queensland X-ray (to Scott Hocknull). The funders had no role in study design, data collection and analysis, decision to publish, or preparation of the manuscript.

### Grant Disclosures

The following grant information was disclosed by the authors:
An Australian Government Research Training Program Scholarship.
The Paleontological Society.
An International Society of Biomechanics Matching Dissertation Grant.
An Australian Research Council DECRA Fellowship: DE120101503.
The donation of CT scan time and technical assistance by Queensland X-ray.

### Competing Interests

John Hutchinson is an Academic Editor for PeerJ.

### Author Contributions

- Peter J. Bishop conceived and designed the experiments, analysed the data, contributed reagents/materials/analysis tools, prepared figures and/or tables, authored or reviewed drafts of the paper, approved the final draft.

- Scott A. Hocknull conceived and designed the experiments, analysed the data, contributed reagents/materials/analysis tools, authored or reviewed drafts of the paper, approved the final draft.
- Christofer J. Clemente conceived and designed the experiments, analysed the data, contributed reagents/materials/analysis tools, authored or reviewed drafts of the paper, approved the final draft.
- John R. Hutchinson conceived and designed the experiments, analysed the data, contributed reagents/materials/analysis tools, authored or reviewed drafts of the paper, approved the final draft.
- Rod S. Barrett conceived and designed the experiments, analysed the data, authored or reviewed drafts of the paper, approved the final draft.
- David G. Lloyd conceived and designed the experiments, analysed the data, authored or reviewed drafts of the paper, approved the final draft.

## Data Availability

All data and code used are held in the Geosciences Collection of the Queensland Museum, and will be made available upon a request being made to the Collections Manager (geoscience.inquiry@qm.qld.gov.au).

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
