# Peer review of "Cancellous bone and theropod dinosaur locomotion. Part II—a new approach to inferring posture and locomotor biomechanics in extinct tetrapod vertebrates"

_PeerJ, doi:10.7717/peerj.5779_

## Round 0.1 · original submission · Major Revisions

· Academic Editor

Major Revisions

Two of the reviewers recommended minor revisions, and one recommended major revisions. Normally in these cases I let the majority rule. However, Reviewer 1 raises important concerns about generalizing among different individual chickens between Part I and Part II, and about generalizing from chickens to all birds. I am sympathetic to those concerns, having looked at the limb bones of a few hundred chickens and seen firsthand how much they vary. In particular, I find Reviewer 1's final paragraph very convincing (without implying that the suggestions of the other two reviewers are unimportant). I hope that you are able to address those concerns, and I look forward to seeing the revised version in time.

Reviewer 1 ·

Basic reporting

I have no concerns about the basic reporting of the manuscript.

Experimental design

It is not clear to me why the preferred workflow for determining posture should require knowledge of bone architecture PLUS centre of mass PLUS segment masses PLUS muscle geometry PLUS muscle activations PLUS material properties etc….all of which will introduce errors when applied to fossil material. Why have the authors not at least attempted to correlate trabecular architecture directly to midstance posture or whatever behavioural parameters they wish to predict in extinct taxa? As on lines 182-184, the authors are supposedly aiming for “fewer assumptions (i.e. model simplicity)”. Whilst they are undoubtedly an impressive undertaking, these models are far from simple, such that we have ended up in situation where we’ve validated the protocol on a n=1 chicken (which is suboptimal itself, see below) and then applied it to only two fossils in Part III.

Ln 191. The choice of a chicken to validate the protocol seems unjustified. Firstly, this is not the same chicken as from Part I, so we actually have no idea what this particular individual’s trabecular fabric is. Secondly, the Part II chicken wasn’t able to be dissected, so the muscle-specific maximum forces had to be scaled from the literature from a different strain. Thirdly, the fabric orientation of the Part I chicken was not able to be quantified because they were so small that the continuum assumption is invalidated, so we can’t quantitatively compare chicken against chicken.

Ultimately, this means the authors cannot test whether this technique successfully predicts the midstance posture of a chicken. Instead, max. principal stress orientation is compared to the mean for all birds in Part I. So effectively we’ve shown that this protocol can predict a bird-like posture for a chicken. But what I want to know is: can the method predict a chicken-like posture for a chicken? Especially because the supplementary manuscript attached on bird kinematics shows (Bishop et al. PLoS ONE, Fig 1D) that there is a distinct scaling relationship in modern bird posture with leg length.

In order for this technique to be truly ‘validated’, we need to know whether it can successfully distinguish between the mid-stance posture of, say, an ostrich and chicken on the basis of trabecular fabric. If the technique cannot distinguish between two modern taxa with different postures or locomotor repertoires (with all the luxury of associated soft tissue data), does it stand a chance with fossil species?

A further advantage with an ostrich or suchlike is that the authors could then directly compare max. principal stress results from the FEA to the Quant3D within the VOI spheres. And then have a more detailed understanding of where the stress and fabric orientation are agreeing/disagreeing, which would get around the problem highlighted by the authors in Lns 732-734.

Lns 731 – This needs further justification, at least some references. Presumably this is why the authors chose model 8 as their preferred model, over 7? This decision also needs further discussion. It seems like there is very little difference in angular deviation within the femoral head between postures 7 and 8, yet 8 is considerably worse within the distal condyles. Why favour 8, rather than trying to minimise the sum of deviation across the two regions?

Ln 812 – “worst to best”. Are the actual joint angles associated with models 1-8 documented anywhere? Why were these 8 chosen out of all the potential options? Why only 8? And when did the authors decide that convergence had been reached? On the basis of what criteria?

838-840 – Do you have any a priori hypotheses about why you would expect sigma1 to match the fabric orientation in this region, whereas correspondence to sigma3 is only discussed in relation to the femur.

Why is no stereoplot for the tibiotarsal results included?

Ln 1042-1043. I disagree. Part I clearly shows that the methodology fir quantifying fabric orientation breaks down both at small and large scales.

Ln 1075 – This is only the case because the authors have selected model 8 as their ‘optimal’. Model 7 seems equally feasible, and would negate this statement.

Validity of the findings

Ln 923 – “the validity of the approach”. I disagree, I do not think the authors have adequately examined the validity of the approach on a modern theropod, because we don’t know what the trabecular architecture of the modern theropod actually is. What if the chicken fabric orientation could be plotted in the stereoplot space, and it sat at the extremities of space occupied by modern birds? Suddenly the correspondence between fabric orientation and stress direction would not be convincing. I do not think comparing stress orientation to a mean fabric orientation for birds is valid.

Lns 738-739 summarise my concerns about the endeavour in its current form. The authors state “it has been shown that birds as a whole appear to demonstrate a largely consistent pattern of cancellous bone architecture in the femur, tibiotarsus and fibula”. Yet the attached Bishop et al. PLoS manuscript shows that the same birds adopt different levels of crouch at midstance. So, in modern ground-dwelling theropods, posture changes but trabecular bone stays the same. I therefore fail to understand how the present attempt to predict posture from trabecular bone can be justified.

·

Basic reporting

The first three criteria are all met. However, there are several places in the methods, results, discussion, and figure captions where the reader is directed to parts I and III of the series of papers on this project. I would prefer to have this information immediately available, especially the observed bone trajectories to compare with the computed compressional stress trajectories.

Experimental design

The paper meets all four criteria here.

Validity of the findings

All four criteria are met here.

Additional comments

The present text was a much more manageable read compared to Part I and its 44 figures. It is for the most part well written, but I do have some comments and suggestions.

At many places in the text, especially the introduction, there are what appear to be excessively long lists of citations of papers supporting the authors’ claims. These ought to be pared back.

Line 113: I was amused the specific phrase “normal DAILY activity”. Does this mean the results of the study are not relevant to nocturnal animals?

Lines 164-166: another epic citation list. Do you really need ALL of them?

Line 587: Equation (3): at first glance I was rather mystified by this cryptic expression with a square root within a square root. I spent a few minutes deriving it for myself, and then it seemed very obvious. I think for less mathematically inclined readers the authors might want to include a brief derivation of this expression.

Line 731: the authors state “... since hip angles are presumed to be more important for determining overall posture in bipeds”. This sounds like it might be true, but I think a demonstration or justification of this statement ought to be made. I think a graphical one would be good to see. This could show the large angular displacement of the foot for a small angular displacement of the femur.

Lines 905-910: I think more than a qualitative assessment of the degree of correspondence between stress trajectories is needed. Some sort of statistical measure is needed.

Paragraph I in II.5.2: both the abstract, introduction and the title of this section refer to the applicability of the method and results of the present study to extinct taxa. However, the only mention of this topic after the introductory remarks is one sentence in this first paragraph, and nowhere else. I was looking forward to what the authors would say on this matter, but was disappointed. I suspect I have to wait until I read Part III for the gripping conclusion.

Figure 4: Part A: It was nice to see the decreasing angular deviations between Ϭ1 and u1 with iterations of the modelling process, but I would like more explanation for the sudden increase in deviation for the medial femoral condyle at stage 8. Part B: A sequence of full lateral views of the 8 steps of the skeletal model from the initial neutral pose to the solution pose would be much more informative than the few obscured stick views currently shown.

Figure 5: I would really like to see illustrations of the original (CT derived) bone trajectories side-by-side with the computed stress trajectories. I don’t think all the views are necessary here, just a select view to show the typical results. For parts I and J some spherical statistics would be very useful to quantify the degree of alignment of the various vectors. Also, there is no explanation of the what the different colours indicate in the caption.

Figure 6: I don’t think ALL these examples are needed. Reduce the number and include the original bone trajectories for readers to make direct comparisons.

Figure 7: Include the original bone trajectories for readers to make direct comparisons.

·

Basic reporting

The writing is clear, although less coherent in places than in the first contribution. The manuscript is technically sound. It works as a self-contained study but is well-integrated with the companion manuscripts. Cite a convergence paper by Tseng (2011) to justify the number of nodes and elements you settled on (p. 24).

Experimental design

Wonderful application of inertia relief, and good mathematical approach to trabecular and principal stress correspondences. NOT a necessary revision, but a suggestion for the future: You can improve the leg segment reconstructions with more interpolation or surface smoothing proximally. Currently they're a bit choppy. You're approximating the inertial properties just fine and the error is certainly small, but smoother anatomy is more evidently authentic.

Validity of the findings

Fascinating correlations between tertiary principal stresses and trabeculae, and the deviations are instructive (as with the distal tibiotarsus).

Additional comments

Most comments about the previous manuscript in the series apply to the current one. The attached edited copy has suggestions pertinent to this particular manuscript. Definitely expand Figures 6 and 7to include at least one comparative sub-figure from the first manuscript. The accretion is good to highlight how you're building an integrated case, although it may seem unnecessary when the first manuscript is a few clicks away.

---

## Round 0.2 · Minor Revisions

· Academic Editor

Minor Revisions

This version is much closer to acceptance. The reviewers have a few additional suggestions for improvement, which strike me as fair and constructive. Please be diligent in addressing the reviewers' concerns, and I will look forward to seeing your manuscript move forward.

Reviewer 1 ·

Basic reporting

No comment

Experimental design

No comment

Validity of the findings

No comment

Additional comments

I am happier with the further justification and caveats the authors have added, although I still have a few minor comments:

- "For one species, this took an estimated 300–350 hours to perform". Could the authors please include this information in the text then? Both as a justification for their small sample size, and to allow readers to judge if technique is suitable for their future needs.

- Admittedly, I have gotten caught up on the extent to which the authors’ protocol can predict a particular (mid-stance) posture. I accept that the authors are instead trying to link trabecular fabric to a ‘characteristic’ posture, which is mentioned in the Intro and Results (ln 860). Thank you for the clarification in the Discussion Ln 977. Please could the authors just add something in the Discussion clarifying that, in the future, we need to figure out what the ‘characteristic posture’ is actually telling us about locomotor biomechanics? It’s nice to calculate this time- and load-averaged characteristic posture, but it is inevitable that people are going to want to apply this technique to say something quite specific about the biomechanical performance of an extinct taxa. What might ‘characteristic posture’ be a useful predictor of?

- Re Ln 764-767: This is fine. But please also reiterate for clarity in the first paragraph of the Results section that the ‘best’ posture represents both the quantitative minimizing of angular deviation AND qualitative visual inspection of the stress trajectories.

- Re Ln 153: I’m still not sure the authors have clarified when convergence was reached. “Until no further improvement was gained” is not repeatable. How do you define ‘further improvement’? Also, I don’t understand how the authors can produce 8 models and decide that convergence had been reached on the final 8th model. Surely you need to overshoot, produce 9-10 models and check that no further improvements can be made.

- Re: authors response "The fabric orientation in the bones of the chickens studied in Part I were quantified via fabric analysis and plotted on steroplots. Moreover, these plots fell well within the region of space occupied by modern birds. We have now reminded the reader of this fact (and more broadly, why a chicken was chosen as our model species) with a new paragraph in the revised Methods section (pages 6-7, lines 197-205)"

This still seems a little odd to me. So we do have some (albeit limited) quantitative data about the chicken fabric orientation from Part 1 to compare the simulation to. But instead of drawing a species-specific comparison (checking the simulated chicken matches a real chicken), the authors are comparing to an all-bird average. But Figure 22 of Part 1 clearly shows the chickens sit off to one side of the all-bird mean for the femoral head.

At least for this region, please can the authors comment on how well stress direction predicted for the chicken corresponds to fabric orientation as quantified in the femoral head of chickens in Part 1? What is the angular deviation in this region?

·

Basic reporting

All four criteria are met, especially number 2.

Experimental design

All four criteria are met.

Validity of the findings

All four criteria are met.

Additional comments

I found this paper to be an improvement on the earlier version. It is a bit long-winded, but this may be a result of having to keep 6 authors happy. I only saw two wording mistakes:

Line 218: "none-to-node"? Do you not mean "node-to-node"?
Line 855: "facies"? Do you not mean "faces" or "facets".

·

Basic reporting

Already-good basic reporting has improved concomitantly with the minor review suggestions.

The methods figures are instructive and vivid. Some figures (especially in the results) can be improved with labelling, legends, and boxes around related images (see suggestions on commented manuscript).

Perhaps I missed something, but why is the pelvis mass so much greater than those of the limb segments? Does it represent the mass of the whole body, minus the hind legs?

Experimental design

Although I agree with other reviewers' vision of an ideal study with vast comparative scope, the authors now better explain their focus on the chicken model and localized parts of bones. The finite element approach is great, with clever touches involving joint forces and soft tissues.

Validity of the findings

Given the paper's stated focus, the findings are valid. Matches of principal stress and fabric fields aren't perfect, but the authors are clear about the exploratory implications of the work.

Additional comments

Detailed suggestions are on the commented manuscript. The authors can consider that some principal stresses may be constraint artifacts reflecting motion in life, the converse of not finding torsion at the tibia mid-shaft seen in-vivo. Bell et al. (2009) found stresses in hadrosaur jaws that suggesting torsion when rigidly constrained, indicating that in life the dentaries actually rotated about their long axis (likely given their articulation the the predentaries).

---

## Round 0.3 · accepted · Accept

· Academic Editor

Accept

Thank you for your diligence in addressing the concerns of the reviewers. I am satisfied with the revised manuscript, and I am happy to accept it for publication in PeerJ.

The decision of whether or not to publish the peer reviews alongside the paper is entirely yours, and will not affect how your paper is handled going forward. However, I encourage you to do so, for several reasons: (1) For this series of papers in particular, the reviewers invested considerable time and effort in providing constructive criticism. Making the reviews public allows the reviewers to receive more credit for their efforts. (2) The exchange of ideas between you and the reviewers is a valuable part of the scientific process and I'd hate to see it lost. (3) Finally, making the reviews public would contribute to the emerging culture of fairness and transparency in editing and peer review.

#